# Robotized indoor phenotyping allows genomic prediction of adaptive traits in the field

Jugurta Bouidghaghen [1,2], Laurence Moreau[3], Katia Beauchêne[4], Romain Chapuis[5], Nathalie Mangel[6], Llorenç Cabrera-Bosquet[1], Claude Welcker [1], Matthieu Bogard [2] & François Tardieu [1] ✉

Breeding for resilience to climate change requires considering adaptive traits such as plant architecture, stomatal conductance and growth, beyond the current selection for yield. Robotized indoor phenotyping allows measuring such traits at high throughput for speed breeding, but is often considered as non-relevant for field conditions. Here, we show that maize adaptive traits can be inferred in different fields, based on genotypic values obtained indoor and on environmental conditions in each considered field. The modelling of environmental effects allows translation from indoor to fields, but also from one field to another field. Furthermore, genotypic values of considered traits match between indoor and field conditions. Genomic prediction results in adequate ranking of genotypes for the tested traits, although with lesser precision for elite varieties presenting reduced phenotypic variability. Hence, it distinguishes genotypes with high or low values for adaptive traits, conferring either spender or conservative strategies for water use under future climates.

Breeding for the improvement of crop resilience is increasingly necessary for the sustainability of cropping systems and for food security in the context of climate change and growing population[1,2]. Most current breeding schemes are based on yield measurement of thousands of genotypes grown under diverse environmental scenarios, assisted by genomic selection that allows yield prediction for many thousands of untested genotypes based on their genomic information[3,4]. In this approach, the measurement of other traits is most often limited to crop cycle duration, which defines the growing areas in which resulting genotypes can be grown, and to traits that may jeopardize the commercialization of selected candidates, such as resistance to diseases or quality performance (e.g. oil content or protein content in rape seed and wheat, respectively)[5]. However, in the context of climate change, other traits that affect light interception, plant development, transpiration and growth are important for predicting, via statistical or crop models, the suitability of genotypes to future environmental conditions[6–8]. Furthermore, a recent analysis of maize genetic progress suggests that physiological traits involved in plant response to heat and drought, such as leaf growth rate or stomatal conductance, have not been improved over the last 60 years of maize selection[9]. Yield was improved via other traits such as the fine-tuning of phenology and the constitutive increase of grain number, but physiological adaptive traits are still a potential reservoir of interesting alleles for climate change[9].

The progress of high-throughput phenotyping now allows one to measure physiological traits for hundreds of genotypes. Robotized indoor phenotyping platforms allow estimation, with typical

[1]LEPSE, Univ Montpellier, INRAE, Montpellier, France. [2]ARVALIS, Chemin de la côte vieille, Baziège, France. [3]GQE-Le Moulon, INRAE, Université Paris-Sud, CNRS, AgroParisTech, Université Paris-Saclay, Gif-sur-Yvette, France. [4]ARVALIS, 45 Voie Romaine, Ouzouer-Le-Marché, Beauce La Romaine, France. [5]DIA-SCOPE, Univ Montpellier, INRAE, Montpellier, France. [6]ARVALIS, Station de recherche et d'expérimentation, Boigneville, France. ✉e-mail: francois.tardieu@inrae.fr

time definitions of some minutes to one day, of traits that underlie the genetic variability of leaf area and their responses to environmental conditions, e.g. leaf expansion rate, leaf width, phyllochron and leaf number[10–13]. They also allow estimation of traits controlling transpiration, e.g. stomatal conductance[14] and those controlling plant architecture, e.g. the vertical distribution of leaf area and the azimuthal distribution of leaves along the stem, with good heritability[15]. Then, light interception, transpiration, and radiation use efficiency can be simulated in virtual field canopies, which reproduce 3D plants characterized in the indoor platform[15–17]. Field phenotyping also allows measuring leaf area at several dates, for hundreds of genotypes in different fields characterized by measured environmental conditions[18–20]. This can result in the estimation of intercepted light in the same fields and, via model inversion, of leaf area and plant architecture[21–23].

However, the use in breeding of these physiological and growth-related traits faces the difficulty of their high sensitivity to environmental conditions, resulting in large genotype x environment interactions[24–27]. This difficulty is not limited to the extrapolation of trait values from indoor to field conditions: most of these traits also largely vary between fields depending on environmental conditions, making difficult the prediction of traits in one field from those measured in another field[24,28,29]. The relationship between these traits and yield is also highly depending on environmental scenarios[30,31]. Consequently, physiological adaptive traits have not been considered per se in breeding programs[9,32].

The recent development of speed breeding in controlled conditions may offer new opportunities for selection strategies involving plant traits. Speed breeding reduces the duration of each generation by setting environmental conditions favouring rapid development, thereby allowing up to eight generations of selection per year[33,34]. Yield and agronomic traits like disease resistance are predicted based on genomic information at each generation, while a full phenotyping of the most promising genotypes is carried out in the field after some generations[35]. However, this approach also potentially includes, in breeding schemes, other traits measured indoor for training a prediction model used in genomic selection, and phenotyped for selected candidates after a few generations. For instance, in wheat, Watson et al.[36] performed speed breeding involving the length of flag leaves and ear length, in addition to yield. Conditions for the use of speed breeding in our case are that physiological adaptive traits translate from indoor conditions to the field, and are accurate enough to make it feasible to implement rapid cycling based on indoor phenotyping and genomic prediction. Three panels of maize hybrids were used to test these conditions (Table 1, Supplementary Table 1): a 'diversity panel' with 246 hybrids[31], a 'genetic progress panel' with a historical series of 56 commercial hybrids[9] and a 'recent hybrids panel' with 86 commercial hybrids marketed from 2008 to 2020 (most indoor measurements on 20 contrasting hybrids, Supplementary Data 1 and Supplementary Table 2).

In this work, we first show that genotypic values of traits measured indoor closely correlate with those in the field, either directly or via modeling (Table 2). We then show that, although absolute trait values differ if measured indoor or in the field, they still follow common trends in response to environmental conditions, and can be inferred by using an ecophysiological model. Finally, we examine to what extent measurements in indoor platforms can serve to train statistical prediction models that estimate genotypic values of traits based on genomic information only (Table 2).

## Table 1 | Summary of variance components and genomic heritability of considered traits

| Trait | Unit | Panel | # Hyb | # Rep | Mean value | $h_g^2$ | $\sigma_g^2$ | $\sigma_a^2$ | $\sigma_d^2$ | $\sigma_e^2$ |
|---|---|---|---|---|---|---|---|---|---|---|
| Leaf appearance rate (LAR) | Leaf/ day$_{20 °C}$ | Diversity panel | 246 | 11 | 0.251 | 0.63 | 1.4E-04 | 1.1E-04 | 3.4E-05 | 8.1E-05 |
| | | Genetic progress panel | 56 | 7 | 0.265 | 0.63 | 1.7E-04 | 9.2E-05 | 7.4E-05 | 9.9E-05 |
| | | Recent hybrids panel | 50 | 3 | 0.262 | 0.56 | 4.8E-05 | 2.5E-05 | 2.3E-05 | 3.7E-05 |
| Vegetative phase duration | Days$_{20 °C}$ | Diversity panel | 246 | 12 | 68.22 | 0.82 | 4.25 | 3.66 | 0.59 | 0.95 |
| | | Genetic progress panel | 56 | 7 | 63.34 | 0.71 | 7.39 | 5.20 | 2.19 | 3.05 |
| | | Recent hybrids panel | 60 | 9 | 65.02 | 0.68 | 2.11 | 1.24 | 0.86 | 0.99 |
| rh$_{PAD}$ (relative height at 50% of leaf area) | Unitless | Diversity panel | 246 | 11 | 0.308 | 0.74 | 7.8E-04 | 6.3E-04 | 1.4E-04 | 2.7E-04 |
| | | Genetic progress panel | 56 | 7 | 0.279 | 0.69 | 1.5E-03 | 9.9E-04 | 4.9E-04 | 6.8E-04 |
| | | Recent hybrids panel | 20 | 3 | 0.360 | 0.54 | 1.1E-03 | 5.3E-04 | 5.2E-04 | 9.1E-04 |
| Stomatal conductance (gs$_{max}$) | mmol/m$^2$/s | Diversity panel | 246 | 11 | 108.4 | 0.48 | 57.2 | 38.4 | 18.8 | 61.2 |
| | | Genetic progress panel | 56 | 7 | 119.8 | 0.53 | 61.77 | 31.35 | 30.42 | 54.02 |
| | | Recent hybrids panel | – | – | – | – | – | – | – | – |
| Leaf expansion rate (LER) | cm$^2$/day$_{20 °C}$ | Diversity panel | 246 | 11 | 134.6 | 0.61 | 107.6 | 76.2 | 31.3 | 69.0 |
| | | Genetic progress panel | 56 | 7 | 163.9 | 0.62 | 343.8 | 215.0 | 128.8 | 211.5 |
| | | Recent hybrids panel | 20 | 3 | 146.9 | 0.54 | 221.6 | 110.0 | 111.5 | 185.5 |
| Leaf area index (LAI) | Unitless | Diversity panel | – | – | – | – | – | – | – | – |
| | | Genetic progress panel | 56 | 7 | 3.65 | 0.66 | 0.23 | 0.15 | 0.08 | 0.13 |
| | | Recent hybrids panel | – | – | – | – | – | – | – | – |

#Hyb, number of hybrids; for the recent hybrids panel, it is defined by the number of hybrids in the considered fields or in the indoor experiment. #Rep, number of independent values calculated for the considered trait. $h_g^2$, genomic heritability (narrow-sense, see Methods), $\sigma_g^2$, total genetic variance. $\sigma_a^2$ and $\sigma_d^2$, variances explained by additive and dominance relationship matrices[72], respectively. $\sigma_e^2$, residual variance. For estimations per experiment, see Supplementary Table 3.

**Table 2 | Summary of correlation and genomic prediction results for each trait**

| Trait | Method indoor[a] | Method field[b] | Experiment | # Hyb | Observed genotypic value | | | Genomic prediction (genotypic means over experiments) | | | | | |
|---|---|---|---|---|---|---|---|---|---|---|---|---|---|
| | | | | | r (BLUEs correlation)[c] | $r_g$ (genetic correlation)[d] | Eff[e] | # Hyb | r (cross-validation)[f] | Acc (cross-validation)[g] | # Hyb | r (external validation)[h] | Acc (external validation)[i] |
| Leaf appearance rate (LAR) | Time course | Time course | Indoor 2 VS Field 6 | 44 | 0.73 ± 0.07 | 0.69 ± 0.08 | 0.70 ± 0.08 | 302 | 0.58 ± 0.09 | 0.74 ± 0.06 | 50 | 0.53 ± 0.10 | 0.71 ± 0.08 |
| | | | Indoor 1 VS Field 3 | 21 | 0.57 ± 0.16 | 0.43 ± 0.19 | 0.45 ± 0.19 | | | | | | |
| | | | Field 5 VS Field 6 | 44 | 0.71 ± 0.08 | 0.65 ± 0.09 | 0.67 ± 0.09 | | | | | | |
| | | | Field 1 VS Field 3 | 26 | 0.49 ± 0.16 | 0.41 ± 0.17 | 0.41 ± 0.17 | | | | | | |
| Vegetative phase duration | Irrelevant | Recorded | Field 5 VS Field 6 | 44 | 0.88 ± 0.04 | 0.77 ± 0.06 | 0.76 ± 0.07 | 302 | 0.84 ± 0.04 | 0.93 ± 0.02 | 60 | 0.71 ± 0.07 | 0.86 ± 0.04 |
| | | | Field 1 VS Field 3 | 53 | 0.69 ± 0.07 | 0.60 ± 0.09 | 0.62 ± 0.09 | | | | | | |
| | | | Field 2 VS Field 3 | 55 | 0.47 ± 0.11 | 0.40 ± 0.12 | 0.41 ± 0.12 | | | | | | |
| Architecture (rh$_{PAD}$ / ALA) | 3D imaging | UAV imaging | Indoor 2 VS Field 5 | 56 | 0.77 ± 0.06 | 0.66 ± 0.08 | 0.66 ± 0.08 | 302 | 0.65 ± 0.08 | 0.75 ± 0.06 | 20 | 0.42 ± 0.20 | 0.57 ± 0.16 |
| | | | Indoor 1 VS Field 4 | 18 | 0.58 ± 0.17 | 0.45 ± 0.20 | 0.44 ± 0.21 | | | | | | |
| | | | Indoor 1 VS Field 2 | 18 | 0.60 ± 0.17 | 0.50 ± 0.19 | 0.49 ± 0.20 | | | | | | |
| | | | Field 4 VS Field 2 | 18 | 0.50 ± 0.19 | 0.41 ± 0.21 | 0.41 ± 0.21 | | | | | | |
| Stomatal conductance (gs$_{max}$) | Model inversion | Not measurable at HTP | Indoor | – | – | – | – | 302 | 0.56 ± 0.09 | 0.81 ± 0.05 | – | – | – |
| Leaf expansion rate (LER) | Time course | | Indoor | – | – | – | – | 302 | 0.76 ± 0.06 | 0.92 ± 0.02 | 20 | 0.34 ± 0.21 | 0.46 ± 0.19 |
| Leaf area index (LAI) | Modeling | UAV imaging | Indoor 2 VS Field 5_WW | 51 | 0.64 ± 0.09 | 0.55 ± 0.10 | 0.53 ± 0.10 | – | – | – | – | – | – |
| | | | Indoor 2 VS Field 5_WD | 51 | 0.44 ± 0.11 | 0.38 ± 0.12 | 0.37 ± 0.12 | – | – | – | – | – | – |

[a,b]Method for trait measurement indoor and in field, respectively. HTP, high-throughput phenotyping. [c]Correlation between genotypic values (BLUEs) for each couple of experiments. [d]Genetic correlation between experiments assessed using a multivariate mixed model[38,39]. [e]Efficiency of indirect selection, (case of an indirect phenotypic selection based on trait observed values in a given experiment, indoor or in a field), calculated as the accuracy of indirect selection divided by the square root of trait genomic heritability in the target field experiment[52]. [f]Correlation between G-BLUP predicted values and measured values in a cross-validation scheme with diversity and genetic progress panels. [g]Prediction accuracy of genomic selection, calculated as [f]divided by the square root of trait genomic heritability. [h]External validation: Correlation between G-BLUP predicted values (with training on diversity and genetic progress panels) and measured values in recent hybrids panel. [i]Prediction accuracy, calculated as [h]divided by the square root of trait genomic heritability. "–": Standard error (SE) estimates[53] are shown after the ± symbol. For details, see Supplementary Tables 4 and 5.

## Results

### Traits measured indoor correlated with those in the field, depending on categories of traits

A genetic approach based on indoor trait measurements requires that the latter are genetically correlated to measurements of the same traits in the field. However, such comparison is not always possible, because robotized indoor phenotyping can measure traits that would be impossible, or very tedious, to measure in the field, such as stomatal conductance or the 3D leaf distribution on the plant stem. Conversely, some traits measured indoor are largely irrelevant to the field, in particular those performed on whole canopies. Hence, comparisons of the genetic variability of trait values obtained indoor and in the field face different levels of difficulty depending on traits. We focused our study on traits that are heritable and have a direct impact on biomass accumulation (Table 1 and Supplementary Table 3). They present contrasting 'phenotypic distances'[37] between indoor and field measurements, thereby causing different degrees of complexity.

Leaf appearance rate (LAR) represents the simplest case, as it is measured with the same protocol indoor and in the field. Its genomic heritability was 0.63 in the diversity and genetic progress panels (Table 1). In the 'recent hybrid panel' (Fig. 1 and Supplementary Tables 2 and 4), correlations between genotypic values indoor and in the field (Fig. 1a and Table 2) were measured either via correlations between BLUEs estimated values or via genetic correlations assessed with a multivariate mixed model[38,39]. As expected[40], genetic correlations were lower than correlations between BLUEs, but were still

significant ($p$-value < 0.02). In both cases (Table 2), they were slightly higher than those between one field and another field (Fig. 1b; $r = 0.57$, $n = 21$, $p$-value = 0.007 and $r = 0.49$, $n = 26$, $p$-value = 0.011, respectively, for correlations between BLUEs). The latter are considered here as a benchmark for evaluating the quality of translation from indoor to field experiments. Importantly, the ranking of hybrids and their distribution in highest and lowest quartiles were essentially conserved between indoor and field conditions, a necessary condition for breeding (Supplementary Table 4). Furthermore, these correlations and rankings were similar to those between fields for the duration of the vegetative phase, a trait that is commonly measured in breeding programmes (Fig. 2a, b).

Plant architecture is a more difficult case because its measurement relies on different principles in indoor vs field experiments (Fig. 3, Table 2, and Supplementary Tables 2 and 4). The architectural trait considered indoor (rh$_{PAD}$) was derived from 3D reconstructions of individual plants, via the difference in altitude between the top of the plant and the point where half of leaf area is reached, normalized by plant height[15]. This trait is closely related to light interception by a canopy[15] and had high heritability (Table 1). It cannot be measured in the field, where 3D reconstruction of individual plants cannot be

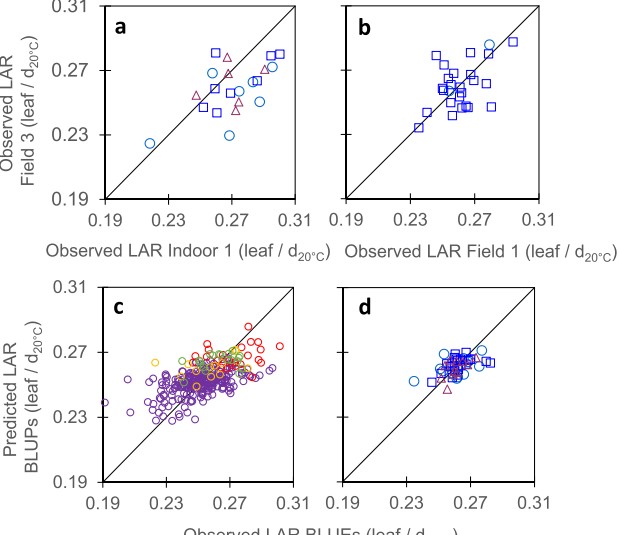

**Fig. 1 | Leaf appearance rate (LAR) translated from platform to field, and could be inferred via genomic prediction. a** Correlations between genotypic values measured indoor and in a field. **b** Correlations between one field and another field were similar to those between indoor and a field. **c** Comparison of observed mean genotypic values and mean predicted values (G-BLUPs) in a 5-fold cross-validation scheme with 10 iterations. **d** Comparison of observed mean genotypic values and predicted values in the independent dataset, with observed values originating from data of **a**, **b** (BLUEs) and G-BLUP model calibration made using dataset of **c**. In **a**, **b** and **d** light blue circles, mid-early hybrids (G2), dark blue squares, intermediate hybrids (G3), red triangles, mid-late hybrids (G4). In **c**, purple empty circles, diversity panel; red and yellow empty circles, genetic progress panel, hybrids released before 1980 and 2000, respectively; green empty circles, hybrids released after 2000. In **a**, $r = 0.57$ (95% CI = 0.19–0.81), $n = 21$, df = 19, $p$-value = 0.007, CV$_{RMSE}$ = 7.7%. In **b**, $r = 0.49$ (95% CI = 0.12–0.74), $n = 26$, df = 24, $p$-value = 0.011, CV$_{RMSE}$ = 5.3%. In **c**, $r = 0.58$ (95% CI = 0.50–0.65), $n = 302$, df = 299, $p$-value < 2.2E-16, CV$_{RMSE}$ = 5.2%. In **d**, $r = 0.53$ (95% CI = 0.30–0.71), $n = 50$, df = 48, $p$-value = 6.3E-05, CV$_{RMSE}$ = 2.8%. Significance of the correlation coefficients was tested using two-sided $t$-test. For spearman correlation of ranks (rho) and other statistics, see Supplementary Tables 4 and 5. Source data are provided as a Source Data file.

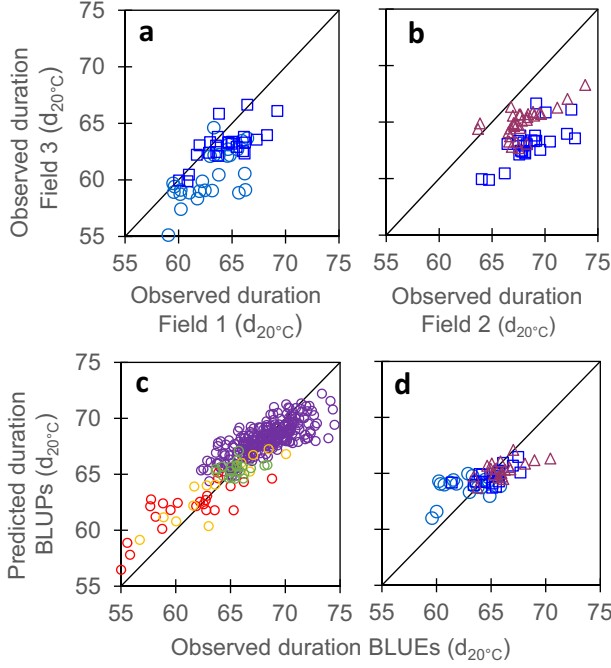

**Fig. 2 | The duration of the vegetative phase, a trait measured in most breeding schemes, was not better related between different fields than other traits compared indoor and in the field in Figs. 1–4. a**, **b** Comparison of observed values in 3 field experiments. **c** Comparison of observed mean genotypic values and mean predicted values (G-BLUPs) in a 5-fold cross-validation scheme with 10 iterations. **d** Comparison of observed mean genotypic values and predicted values, in the independent dataset. Observed values originated from data of **a**, **b** (BLUEs) and G-BLUP model calibration was performed using dataset of **c**. In **a**, **b** and **d**, light blue circles, mid-early hybrids (G2), dark blue squares, intermediate hybrids (G3), red triangles, mid-late hybrids (G4). In **c**, purple empty circles, diversity panel, red and yellow empty circles, genetic progress panel, hybrids released before 1980 and 2000, respectively; green empty circles, hybrids released after 2000. In **a**, $r = 0.69$ (95% CI = 0.52–0.81), $n = 53$, df = 51, $p$-value = 9.4E-09, CV$_{RMSE}$ = 4.4%. In **b**, $r = 0.47$ (95% CI = 0.23–0.65), $n = 55$, df = 53, $p$-value = 0.0003, CV$_{RMSE}$ = 7%. In **c**, $r = 0.84$ (95% CI = 0.81–0.89), $n = 302$, df = 300, $p$-value < 2.2E-16, CV$_{RMSE}$ = 2.7%. In **d**, $r = 0.71$ (95% CI = 0.56–0.82), $n = 60$, df = 58, $p$-value = 1.5E-10, CV$_{RMSE}$ = 2.5%. Significance of the correlation coefficients was tested using two-sided $t$-test. For rho and other statistics, see Supplementary Tables 4 and 5. Source data are provided as a source Data file.

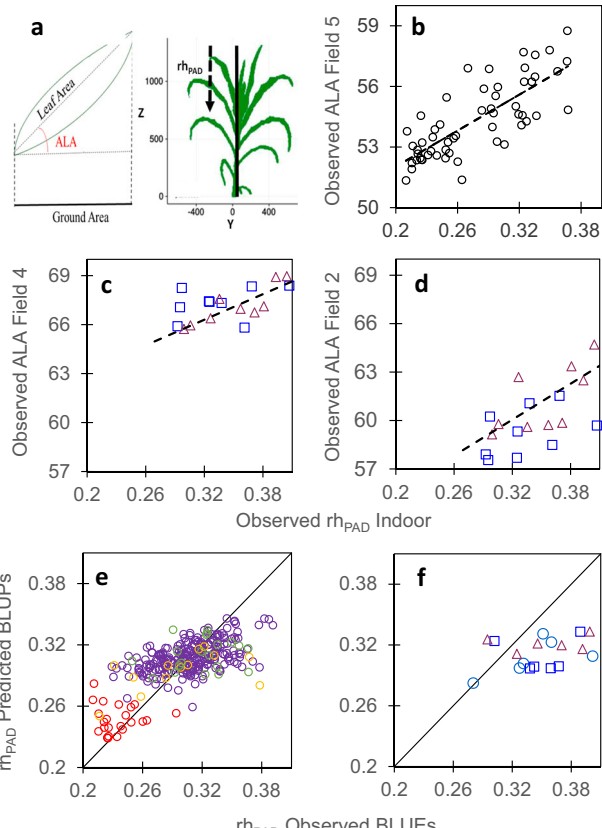

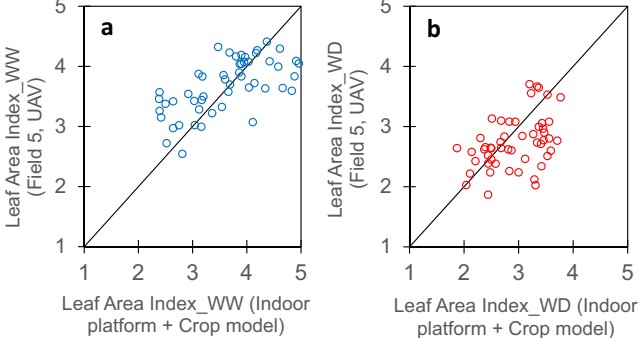

**Fig. 4 | Genotypic values of leaf area index (LAI) measured in the field were correlated with values of LAI derived via a crop model taking into account traits measured indoor and environmental data measured in the considered field. a** well-watered; **b** water-deficit. Field values (on *y* axis) were obtained at flowering time from UAV images via inversion of the model PROSAIL[41,42]. Values on *x* axis were obtained via the crop model of Lacube et al.[11], fed with measured values of (i) environmental conditions in the same field, (ii) genotypic values measured in the platform for leaf appearance rate (LAR), maximum leaf growth rate (LER), responses of leaf growth rate to vapor pressure deficit (VPD) and soil water potential, and final leaf number per plant. Each point, one genotype. In **a**, *r* = 0.64 (95% CI = 0.45–0.78), *n* = 51, df = 49, *p*-value = 3.6E-07, CV$_{RMSE}$ = 15.7%. In **b**, *r* = 0.44 (95% CI = 0.18–0.64), *n* = 51, df = 49, *p*-value = 1.3E-03, CV$_{RMSE}$ = 19.6%. Significance of the correlation coefficients was tested using two-sided *t*-test. For rho and other statistics, see Supplementary Table 4. Source data are provided as a Source Data file.

**Fig. 3 | Plant architecture translated from platform to field, and could be inferred via genomic prediction. a** Schematic representation of average leaf inclination angle (ALA) in the field, estimated from UAV images at flowering time, via inversion of the model PROSAIL[42,75] and rhPAD measured indoor as the relative altitude, from the top of the plant, where 50% of leaf area is reached[15]. **b, c** and **d** Correlations between rhPAD (indoor) and ALA (Fields 5, 4 and 2, respectively) for the genetic progress panel (**b**) and the recent hybrids panel (**c, d**). **e** Comparison of observed mean genotypic values and mean predicted values (G-BLUPs) in a 5-fold cross-validation scheme with 10 iterations for rhPAD. **f** Comparison of mean geno-typic values (BLUEs) and predicted values (G-BLUPs) in the independent dataset. In **b, c** and **f**, light blue circles, mid-early hybrids (G2), dark blue squares, intermediate hybrids (G3), red triangles, mid-late hybrids (G4). In **d**, purple empty circles, diversity panel; red and yellow empty circles, genetic progress panel, hybrids released before 1980 and 2000, respectively; green empty circles, hybrids released after 2000. In **b**, *r* = 0.77 (95% CI = 0.64-0.86), *n* = 56, df = 54, *p*-value = 2.85E-12. In **c**, *r* = 0.58 (95% CI = 0.15-0.82), *n* = 18, df = 16, *p*-value = 0.012. In **d**, *r* = 0.60 (95% CI = 0.18-0.83), *n* = 18, df = 16, *p*-value = 0.009. In **e**, *r* = 0.65 (95% CI = 0.59-0.72), *n* = 302, df = 297, *p*-value < 2.2E-16, CV$_{RMSE}$ = 9.4%. In **f**, *r* = 0.42 (95% CI = −0.02-0.73), *n* = 20, df = 18, *p*-value = 0.06, CV$_{RMSE}$ = 15.6%. Significance of the correlation coefficients was tested using two-sided *t*-test. For rho and other statistics, see Supplementary Tables 4 and 5. Source data are provided as a Source Data file.

performed. Conversely, drone imaging in the field results in the calculation of a related trait, the Average Leaf inclination Angle (ALA), derived from the inversion of the radiative transfer model 'PROSAIL'[41,42], which takes into account the deviation of light interception efficiency of a given canopy in relation to a standard canopy having the same leaf area. The genotypic values of ALA measured in the field correlated to those of rhPAD measured in a phenotyping platform in an experiment with 56 maize hybrids of the 'genetic progress' panel (Fig. 3b, Field 5, Supplementary Tables 2 and 4). The same applied to 20 hybrids of the 'recent hybrids' panel in two field experiments, with good relationships between rhPAD and ALA (Fig. 3c, d), high heritability of both variables (Supplementary Fig. 1) and good conservation of lowest and highest quartiles (Supplementary Table 4). Notably, ALA values and heritability were sensitive to crop phenological stage

whereas those of rhPAD were more stable (Supplementary Fig. 1). Hence, architectural data collected indoor were, in this case, appropriate for characterizing each genotype in models of light interception, whereas ALA measured in the field would be more complex to use in this context.

In the same way, the leaf expansion rate of individual plants (LER) can only be measured indoor, with good heritability (Table 1)[43]. Corresponding measurements in the field are leaf area or leaf dimensions at given dates, so direct comparisons were not possible. However, we show below that the final width and length of maize 8th leaf matched between indoor and field conditions for the diversity panel. Hence, final leaf dimensions potentially allow indirect calculation of LER in the field[44].

Leaf area index (LAI), a key feature for light interception and transpiration, is defined for a fraction of field canopy (typically 1 m²). Although heritable within a given field, it largely differs between fields in relation to environmental conditions and plant density[45]. It can be measured indoor, but a direct comparison with the field would make no sense because the density and spatial arrangement of plants in indoor experiments make the considered canopy irrelevant to the field[24,46]. Indeed, LAI measured in the field was not correlated to the LAI calculated by considering plant leaf area measured indoor at flowering time, multiplied by the plant density in the corresponding field (Supplementary Fig. 2, *r* = −0.25, *n* = 51, *p*-value = 0.073). This was because environmental conditions and management practices were too different between the greenhouse and the field. Instead, we calculated LAI based on the genotypic values of upstream traits measured indoor (Table 2 and Fig. 4). We compared (i) measured values in the field, obtained via UAV imaging and the inversion of the PROSAIL radiative transfer model[41,42] with (ii) the LAI simulated by a crop model[11]. Model inputs were the genotypic values of four traits measured in indoor platform (LAR, maximum leaf growth rate (LER), responses of leaf growth rate to VPD and soil water potential, and final leaf number), plus plant density and the environmental conditions recorded every hour in the considered field. The correspondence between measured and estimated LAI, tested on the 'genetic progress' panel suggested that this approach is promising in well-watered (WW) condition

($r = 0.64$, $n = 51$, $p$-value = 3.6E-07, Fig. 4a), and even in water-deficit (WD) condition although the indoor platform experiment was performed in WW condition, except for the response of leaf growth rate to soil water potential ($r = 0.44$, $n = 51$, $p$-value = 1.3E-03, Fig. 4b and Supplementary Table 4). Notably, the PROSAIL model inversion allowing LAI estimation in the field always resulted in values lower than 4.5 for all hybrids. When the real LAI was higher, light interception efficiency was close to 100%, so model inversion could not provide LAI values higher than 4.5[47].

Finally, stomatal conductance is a difficult case in which traits cannot be directly measured at high throughput, either in the field or in indoor platforms. Its measurement for one leaf requires between 3 and 15 min, depending on the considered device, making high-throughput measurements impossible. However, it can be indirectly estimated at plant level in platform experiments by inversion of the Penman-Monteith equation, based on measurements of individual plant transpiration, leaf area, light, and VPD[14]. Resulting estimations of whole-plant stomatal conductance were well related to leaf stomatal conductance measured via gas exchange, between well-watered and water deficit treatments (Fig. 5a and Supplementary Table 1), but also between genotypes in the well-watered treatment ($r^2 = 0.54$).

Overall, the ranking of genotypes for leaf appearance rate, plant architecture, and LAI were consistent between field and indoor conditions, thereby opening the way for a prediction of values in the field based on platform information (Table 2 and Supplementary Table 4). This could not be tested for stomatal conductance, for which field measurements cannot be performed and indirect measurements via canopy temperature are not precise enough in non-extreme conditions.

## The differences in absolute values of traits between indoor and fields were accounted for by environmental conditions

Beyond the correlations between genotypic values of traits measured indoor and in the field, it is the absolute values of traits, measured in each experiment, that eventually drive the adaptation of studied genotypes to drought and high temperature. For example, a correct estimation of genotype ranking for LAI has a very small impact on light interception if all genotypes have a LAI higher than 4, whereas the same genotype ranking in a range of LAI from 2 to 4 has a large impact[47]. Meta-analyses showed that phenotypic values differ between controlled conditions and field[24], but they also largely vary from one field to another one[11,24]. Hence, we tested if the difficulty for translating values between two experiments may not be specific to field – platform comparisons, but applies to comparisons between any environment and another one, depending on environmental conditions in each experiment.

This hypothesis was first tested by examining the mean absolute values of maize leaf length and width between field and indoor platforms for the diversity panel in Lacube et al.[44]. A superficial analysis would suggest that the mean dimensions of leaf 8 largely differed between indoor and field conditions, with a mean leaf length of 115 vs 76 cm, respectively, and a mean leaf width of 6.8 cm vs 7.5 cm, respectively (note the inversion of ranking between the two traits). However, leaf dimensions also largely varied among field experiments, from 6.8 to 10 cm for leaf width and from 68 to 102 cm for leaf length (Fig. 6a, b). We showed earlier that leaf width depends on the amount of light intercepted during the growth of the considered leaf[44]. Accordingly, leaf width in the field was linearly related to the cumulated intercepted light ($r = 0.83$, $n = 64$, $p$-value < 2.2E-16), and the same relationship accounted for the difference between experiments in fields and platform (Fig. 6a). In the same way, the large variability of leaf length, in field and controlled conditions, was accounted for by the vapor pressure deficit (VPD) during leaf growth (Fig. 6b, $r = -0.62$, $n = 44$, $p$-value = 6.2E-06), consistent with studies showing a linear effect of VPD on leaf elongation rate[43]. Hence, leaf width and length did

not differ intrinsically between indoor and field conditions: differences were accounted for by the same environmental conditions than those that accounted for differences between one field and another one, and could be calculated via a crop model[11].

A similar case occurred with temperature-dependent traits, such as the duration of the vegetative phase (Fields 1, 2, and 3, Supplementary Table 2) or leaf appearance rate (Fields 1, 3 and PhenoArch, Supplementary Table 2). When expressed in calendar time, these trait values differed greatly between environments (Supplementary Fig. 3a, b), whereas they were consistent if the effect of temperature was taken into account via a model of thermal time[48,49]. Expressed in this way, measured values were similar between field experiments for the duration of the vegetative phase and for LAR (Figs. 2a, b and 1b, respectively) or between a field experiment and an indoor platform experiment (Fig. 1a), although some differences still existed between experiments ($CV_{RMSE}$ = 4.4% and 7% in Fig. 2a, b, $CV_{RMSE}$ = 5.3% and 7.7% in Fig. 1a, b). Among possibilities for explaining such differences in duration of the vegetative phase and LAR, the frequency of field visits was three days on average, but slightly differed between experiments.

Overall, values translation from indoor platforms to field, and from one field to another field, could be carried out for a range of traits by taking into account appropriate environmental variables.

## Measurements in indoor platforms can be used for genomic prediction of traits

High-throughput phenotyping allows characterization of some hundreds of genotypes (at most) whereas many thousands of genotypes are required for breeding[3,4]. In the same way, it would not be feasible to phenotype the offspring at each generation of speed breeding because of the resulting cost and workload[34,36]. Hence, the use of physiological traits in breeding requires one's ability to predict them based on genomic information, as it is the case for yield[50,51]. We have tested this possibility for the traits presented in the former paragraphs. Briefly, we trained a G-BLUP model based on the 246 hybrids of the 'diversity panel' and the 56 hybrids of the 'genetic progress' panel (Supplementary Table 1). This training was performed with the genotypic means (BLUEs over the experiments carried out in Millet et al.[31] and Welcker et al.[9]) of the duration of the vegetative phase, the leaf appearance rate, maximum leaf expansion rate (calculated with two methods based on different assumptions, see "Methods" section), the architectural trait $rh_{PAD}$ and stomatal conductance. Predictions were performed using the genomic information at 440 000 polymorphic SNPs. Prediction accuracies and RMSEs were assessed either with a 5-fold cross-validation (CV1) scheme[28] (random sampling of hybrids using a stratification strategy for respecting the proportions of genetic groups, Supplementary Fig. 4), or with an external validation set made of genotypic trait means estimated in the 'recent hybrids panel' (independent experiments, Supplementary Data 1).

Cross-validation provided good quality of prediction for studied traits, assessed either by the correlation (r) between observed BLUEs and predicted G-BLUPs values, or by the prediction accuracy of genomic selection (Acc), calculated by dividing the correlation coefficient (r) by the square root of trait genomic heritability[52,53] (Table 2). This was the case for leaf appearance rate ($r = 0.58$, $n = 302$, $p$-value < 2.2E-16, $CV_{RMSE}$ = 5.2% in Fig. 1c), leaf expansion rate ($r = 0.76$, $n = 302$, $p$-value < 2.2E-16, $CV_{RMSE}$ = 8.7% in Fig. 7a), $rh_{PAD}$ ($r = 0.65$, $n = 302$, $p$-value < 2.2E-16, $CV_{RMSE}$ = 9.4% in Fig. 3e) and stomatal conductance ($r = 0.56$, $n = 302$, $p$-value < 2.2E-16, $CV_{RMSE}$ = 8.4% in Fig. 5b) (Supplementary Table 5). These values are similar but slightly lower than those for the duration of the vegetative phase in our study ($r = 0.84$, $n = 302$, $p$-value < 2.2E-16, $CV_{RMSE}$ = 2.7% in Fig. 2c), and for yield or flowering time traits in other studies[51,54–56]. Notably, genomic prediction with G-BLUP model performed similarly for the genotypes originating from the two panels, in spite of the difference in structure and origins of these panels (Supplementary Fig. 5) and the fact that measurements

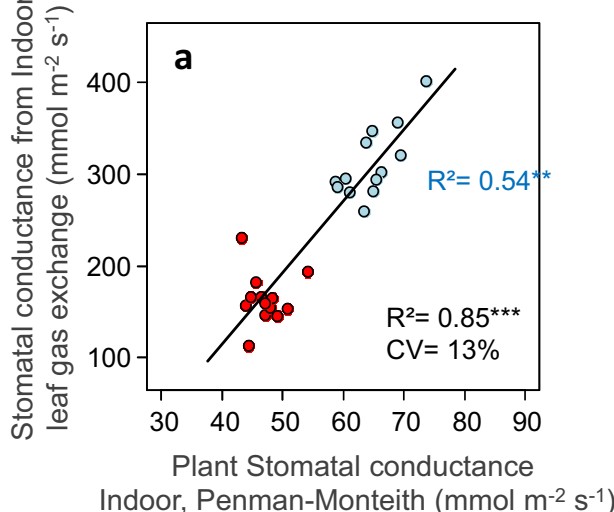

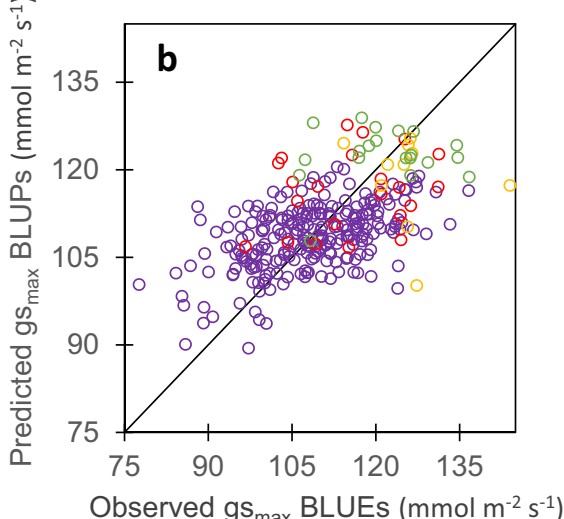

**Fig. 5 | Stomatal conductance can be measured at plant level in an indoor phenotyping platform and predicted from genomic information. a** Comparison between values obtained at leaf level via gas exchange, and at plant level via inversion of the Penman-Monteith equation[14], in well-watered and water deficit treatments. **b** Comparison of observed mean genotypic values and mean (G-BLUP) predicted values in a 5-fold cross-validation scheme with 10 iterations for plant stomatal conductance. In **a** and **b**, each symbol, one genotype; blue, well-watered; red, water deficit. In **a**, black line, linear regression. In **b**, black line is the 1:1 line. In **a**, $r = 0.92$ (95% CI = 0.83–0.96), $n = 26$, df = 24, $p$-value = 2.4E-11, $CV_{RMSE} = 13\%$. In **b**, $r = 0.56$ (95% CI = 0.47–0.63), $n = 302$, df = 293, $p$-value < 2.2E-16, $CV_{RMSE} = 8.4\%$. Significance of the correlation coefficients was tested using two-sided t-test. For rho and other statistics, see Supplementary Table 5. Source data are provided as a Source Data file.

were performed in different experiments. Furthermore, when predicting traits with a PC-BLUP model that is used as predictors the genotypes coordinates on the first five axes of SNP PCoA (Principal Coordinate Analysis) of the panels (Supplementary Fig. 5), the prediction quality decreased when considering individual clouds of points corresponding to each panel (Supplementary Fig. 6 and Supplementary Table 6).

The range of G-BLUP predicted values was expectedly smaller, for all tested traits, than that of observed values. This bias is linked to the fact that the narrow-sense heritability estimated using genomic additive and dominance relationships of studied traits was lower than 1 (0.68–0.82, 0.56–0.63, 0.54–0.62, 0.54–0.74, and 0.48–0.53 for the duration of the vegetative phase, LAR, leaf expansion rate, $rh_{PAD}$ and

stomatal conductance, respectively, Table 1). Hence, the prediction based on genomic information covered a smaller range of values than original data.

The external validation was a more challenging scheme, where we tried to predict the performance of new genotypes evaluated in new independent experiments. Moreover, the 'recent hybrids panel' used here covered smaller ranges of trait phenotypic values than those in the 'diversity' and 'genetic progress' panels considered jointly. Consequently, the comparison of observed vs G-BLUP predicted values led to lower prediction accuracies than in the case of cross-validation ($r$ ranged between 0.34 and 0.71, Table 2). This applied to traits measured indoor (LAR, Fig. 1d, LER, Fig. 7b and $rh_{PAD}$, Fig. 3f) as well as for the duration of the vegetative phase measured in the field (Fig. 2d and Supplementary Table 5), so this problem was not specific of indoor genomic prediction. The external validation using the simple PC-BLUP model resulted in much lower prediction accuracies than that using the G-BLUP model (Supplementary Fig. 6 and Supplementary Table 6). This suggests that G-BLUP predictions captured genetic effects beyond that explained by population structure.

## Discussion

Three conditions can be considered as requirements for traits measured indoor to be used in trait-based selection in a context of climate change. Firstly, traits measured indoor should be genetically correlated to those in fields (regardless of absolute values either indoor or in each field), so indoor breeding is relevant to field conditions. Secondly, the absolute value of indoor traits should translate to that in fields with diverse climate scenarios, either directly or via models. Finally, indoor traits need to be predicted with sufficient accuracy from the genomic information of non-phenotyped genotypes.

The traits presented here satisfied the first condition. Close correlations were observed between the genotypic values of traits measured indoor and in multi-site field experiments. This was the case when the considered trait was measured with similar protocols indoor or in the field, for instance leaf appearance rate or the duration of the vegetative phase. It was also the case when the trait was measured with different methods as in the case of plant architecture. Finally, the integrated trait LAI, which is highly dependent on the plant density and environmental conditions in the considered canopy, required a method involving crop modeling. The correlations observed in these three cases between indoor platform and fields are therefore higher ($r = 0.57$ to 0.77, Table 2) than those reported by Poorter et al.[24] for a set of growth-related traits meta-analysis (median $r = 0.51$). Two reasons may explain this disparity. (i) Traits considered in Poorter's meta-analysis, namely yield, leaf nitrogen concentration and specific leaf area are more integrated than those studied here (except LAI) and, therefore more prone to high genotype x environment interactions and changes in the ranking of genotypes. (ii) The traits studied here had moderate to high heritabilities over experiments, thereby showing a low residual variance resulting from experimental errors. Furthermore, measuring yield, yield components, or leaf area index in phenotyping platforms is probably not relevant because these traits result from cumulative processes over a long period, during which conditions indoor are very different from those in the field. The methods presented here for comparing indoor and field trait values considerably reduced the genotype x environment interaction (GEI) for such integrated traits. For example, a direct comparison of LAI indoor and in the field resulted in a high GEI, without correlation between them. Conversely, the GEI was largely reduced when upstream traits measured indoor (with a low GEI) were combined, via a crop model, with the management practices and environmental conditions in the considered field[37].

The second condition, namely that trait values can translate from indoor conditions to a diversity of fields, was fulfilled for the traits reported here if the differences in environmental conditions were

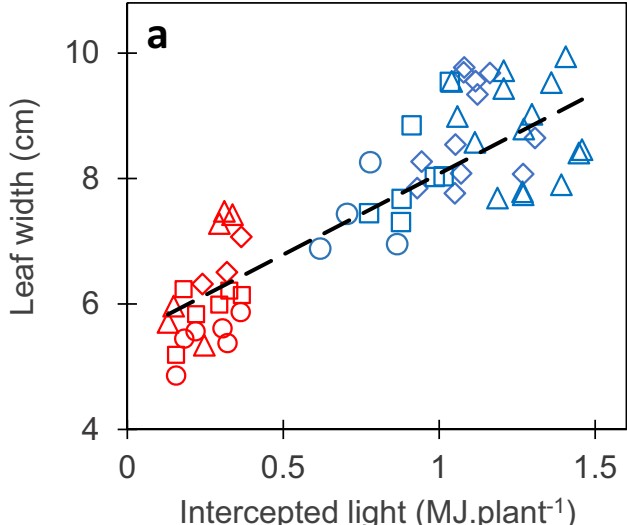

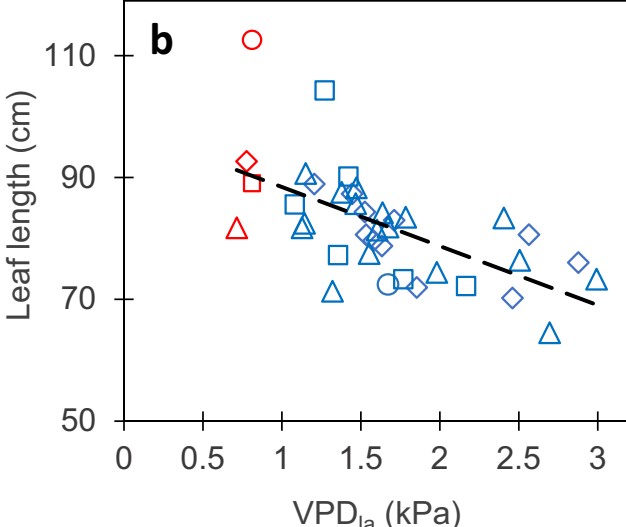

**Fig. 6 | Leaf width and length responded similarly to environmental conditions in fields and indoor platforms. a** Relationship between leaf width and the cumulated light intercepted by plants during leaf widening. **b** Relationship between leaf length and leaf-to-air vapor pressure deficit (VPD$_{la}$: mean of maximum daily values) during leaf elongation. Each point, one experiment and leaf rank. Leaf width and length values of four leaf ranks (8–11, circles, squares, diamonds and triangles, respectively) were corrected for leaf rank so equivalent values for leaf 8 are presented[44]. Blue dots: field, red dots: indoor platform. Black lines, linear regressions. In **a**, $r = 0.83$ (95% CI = 0.73-0.89), $n = 64$, df = 62, $p$-value < 2.2E-16. In **b**, $r = -0.62$ (95% CI = −0.78-0.40), $n = 44$, df = 42, $p$-value = 6.2E-06. Significance of the correlation coefficients was tested using two-sided $t$-test. Source data are provided as a Source Data file.

taken into account, thereby dealing with the GEI via a previously reported model[31].

- Modeling temperature effects allowed consistency between field and platform experiments for leaf appearance rate and the duration of the vegetative phase in this study. This result can be generalized to traits related to the progression of plant development of many species. In particular, germination rate, leaf appearance rate, the reciprocal of the duration of growth of individual leaves and reproductive organs are common to a large range of environments if they are expressed in thermal time[57]. Crop models, based on this result, successfully predict plant phenology in wide ranges of environments[8,30].

- The amount of intercepted light was also needed for other traits to be consistent between experiments. This was the case here for maize leaf width measured indoor and in several fields. Beyond this particular trait, Monteith showed that biomass accumulation is proportional to the cumulated light intercepted by plants[47]. In particular, we showed that, in a series of experiments with maize, the time course of plant biomass largely differed between experiments but was consistent if expressed as a function of intercepted light[16]. Again, crop models based on intercepted light can predict plant biomass accumulation with reasonable accuracy[30,58].

- Plant water status was, in addition, necessary to account for differences in traits related to organ expansive growth (expressed in terms of volume or length). Its effect can be predicted from the cell scale[59] to the organ scale[60–62]. Here, this was the case for leaf length in well-irrigated maize fields, as a function of air VPD. Leaf elongation rate is closely related to a combination of soil water potential and VPD in maize, fescue, or barley[60,63], so our result can probably be extended to other species. Stomatal conductance can also be predicted from a combination of soil water status, evaporative demand, and incident light via functional models involving chemical and hydraulic signals[64]. Crop models that take into account light, soil water content and evaporative demand can predict stomatal conductance and net photosynthesis by simulating physiological processes[65,66], so photosynthesis in controlled conditions can be extended to a range of field conditions[66].

Overall, we confirmed that raw phenotypic traits cannot translate directly from indoor platforms to fields, as reviewed in Poorter et al.[24]. However, taking into account specific environmental conditions allowed this translation for the traits presented here, which depend on one or two environmental conditions. Again, more integrated traits such as leaf area index, grain number or grain yield measured in a platform cannot be directly extended to field via simple relationships as presented in former paragraphs. These traits can be predicted in a range of field conditions based on genotype-specific parameters and environmental conditions measured in the considered fields. This was the case here for leaf area index, but was also the case for grain number and grain yield in a multi-site field experiments, based on a mixed model involving genetic parameters and environmental conditions[31].

The third condition is that traits can be predicted from genomic information. Here, cross-validation based on a large genetic range showed good results (compared to Guo et al.[56], Yuan et al.[55], or Toda et al.[51]), with r ranging from 0.56 to 0.84 for the studied traits (Table 2 and Supplementary Table 5). External validation on the panel of recent hybrid varieties provided less accurate results, but correlations between predicted and observed values still ranged from 0.34 to 0.71 and mostly with significant $p$-values (from 1.5 E-10 to 0.14) and acceptable CV of errors (from 2.5% to 15.6%) (Supplementary Table 5). By using this panel for external validation, we chose the most challenging case, in which one attempts to use a genomic prediction model, trained with a panel with wide genetic variability, to predict elite genotypes that have a reduced phenotypic variability for studied traits. Hence, our results could not be considered as fully satisfactory if the purpose was to rank elite genotypes (Supplementary Table 5). Conversely, the cross-validation in a wider genetic range suggests that genomic prediction may be used for identifying genotypes with high or low genotypic values for studied traits in breeding populations with higher genetic and phenotypic variabilities. This allows the design of ideotypes with contrasting strategies in relation to water and heat stress, namely 'conservative' ideotypes with low stomatal conductance, leaf growth, leaf appearance rate, for stress-prone areas, vs 'spender' ideotypes with highest values for each of these traits.

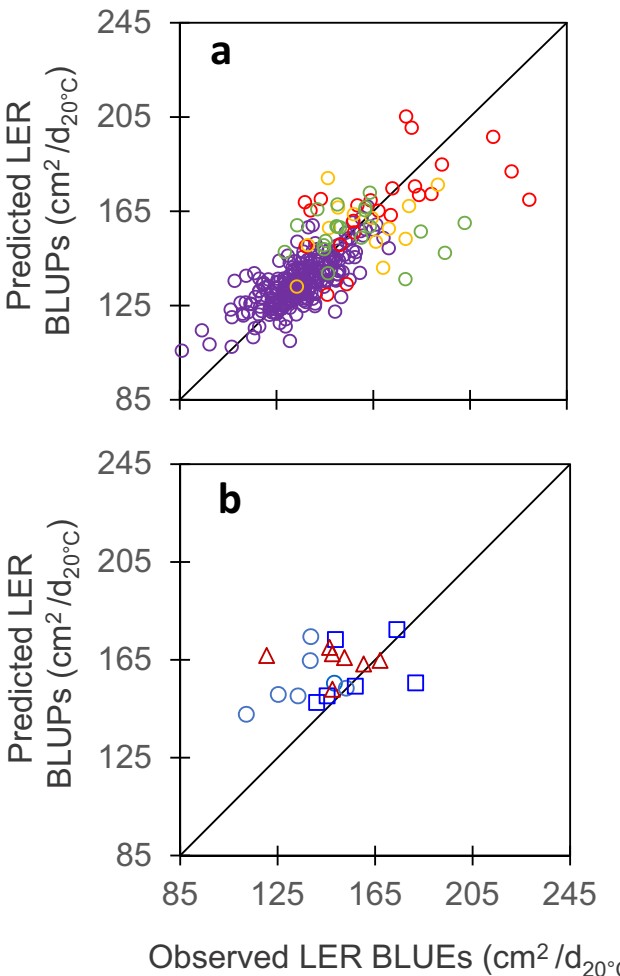

**Fig. 7 | Maximum leaf expansion rate (LER) could be predicted from genomic information via cross-validations in panels with large phenotypic variability but predictions were not accurate for the independent dataset. a** Comparison of observed mean genotypic values and mean (G-BLUP) predicted values in a 5-fold cross-validation scheme with 10 iterations for plant max leaf expansion rate (LER). **b** Comparison of observed mean genotypic values and predicted values for the recent hybrids dataset, with G-BLUP model calibration made using dataset of **a**. LER (Leaf Expansion Rate) was extracted from time courses of leaf area in the platform[14] and determined as the slope of the linear regression between leaf area and thermal time during the period from 24 to 45 $d_{20°C}$. In **a**, purple empty circles, diversity panel; red and yellow empty circles, genetic progress panel, hybrids released before 1980 and 2000, respectively; green empty circles, hybrids released after 2000. In **b**, light blue circles, mid-early hybrids (G2), dark blue squares, intermediate hybrids (G3), red triangles, mid-late hybrids (G4). In **a**, $r = 0.76$ (95% CI = 0.71-0.82), $n = 302$, df = 297, $p$-value < 2.2E-16, $CV_{RMSE} = 8.7\%$. In **b**, $r = 0.34$ (95% CI = −0.12– 0.68), $n = 20$, df = 18, $p$-value = 0.14, $CV_{RMSE} = 14.1\%$. Significance of the correlation coefficients was tested using two-sided $t$-test. For rho and other statistics, see Supplementary Table 5. Source data are provided as a Source Data file.

## Methods

### Genetic materials

Three panels of maize hybrids were used in this study (Supplementary Table 1). First, a diversity panel included 246 hybrids resulting from the cross of a common flint parent (UH007) with 246 dent lines that maximized the diversity in the dent group while keeping a restricted flowering window[31,67]. This panel involved four genetic groups, namely Iodent (39 hybrids), Lancaster (45 hybrids), Stiff-Stalk (55 hybrids), and diverse-dent hybrids (107) consisting in an admixture of the former three groups[67]. Second, a 'genetic progress' panel included 56 highly successful commercial hybrids released on the European market from 1950 to 2015[9]. This panel showed a limited range of maturity classes,

from mid-early (FAO 280) to mid-late (FAO 480), covering the largest growing area in Europe. Finally, a 'recent hybrids' panel included 86 commercial hybrids released from 2008 to 2020, belonging to mid-early to mid-late maturity classes (Supplementary Data 1). Yield data in 30 sites x 2 years per hybrid were available at the beginning of this study (ARVALIS, www.varmais.fr).

### Platform experiments

Platform experiments were performed in PhenoArch, an indoor robotized and image-based phenotyping platform that allows precise measurement of plant architecture, plant phenology and growth, transpiration, stomatal conductance and water use efficiency (https://www6.montpellier.inrae.fr/lepse/Plateformes-de-phenotypage-M3P/Montpellier-Plant-Phenotyping-Platforms-M3P/PhenoArch)[16] hosted at Montpellier Plant Phenotyping Platforms (M3P). The diversity panel was evaluated in four experiments (in spring 2012, 2013, and 2016, and winter 2013) as described in Prado et al.[14]. Three or two plants per hybrid were grown depending on the experiment (Supplementary Table 1). The 'genetic progress' panel was evaluated in four experiments, with most data used here originating from an experiment with seven replicates per hybrid[9]. A subset of 20 hybrids of the 'recent hybrids' panel was evaluated in one experiment during winter 2021, with three replicates per hybrid. All experiments followed an alpha-lattice design, with two levels of soil water content imposed, namely retention capacity (well-watered, soil water potential of −0.05 MPa) and water deficit (soil water potential from −0.3 to −0.6 MPa depending on the experiment). Soil water content in pots was maintained at target values by compensating transpired water three times per day via individual measurements of each plant[16]. Soil water potential was estimated from soil water content based on a water release curve[14]. Air temperature and humidity were measured at six positions in the platform every 15 min. Daily incident photosynthetic photon flux density (PPFD) over each plant within the platform was estimated by combining a 2D map of light transmission, and the outside PPFD measured every 15 min with a sensor placed on the greenhouse roof[16]. The greenhouse temperature was maintained at 25 ± 4 °C during the day and 17 ± 2 °C during the night. Supplemental light was provided either during daytime when external solar radiation was below 300 W m⁻² or to extend the photoperiod by using 400 W HPS Plantastar lamps.

In each experiment, the number of visible leaves of every plant was manually scored weekly during the vegetative phase. Leaf appearance rate (LAR, reciprocal of the phyllochron) was calculated as the slope of the linear relationship between the number of visible leaves and thermal time, during the period from plant emergence to 12-leaf stage. Red-Green-Blue (2056 × 2454) images taken from 13 views (12 side views from 30° rotational difference and one top view) were captured daily for each plant during the night. Plant pixels from each image were segmented from those of the background and used for estimating the whole plant leaf area and fresh biomass[68]. The time courses of leaf area and plant fresh biomass were then fitted individually by using P-spline growth curve models[69]. The architectural trait $rh_{PAD}$ was calculated daily from 3D reconstructions of each plant, based on RGB images at PhenoArch platform[15]. $rh_{PAD}$ index represented the point in the distribution of leaf area along the stem (from the top of the plant, relative to total plant height) where half of the cumulative leaf area is reached. Whole-plant stomatal conductance was calculated over 4 time-periods per day for 20 days for each hybrid plant in PhenoArch platform, via inversion of the Penman–Monteith equation based on transpiration, plant growth, net radiation and VPD collected in the experiment[14]. Its value under saturating light was estimated for each hybrid by combining coupled values of stomatal conductance and incident light observed in all experiments. The maximum leaf expansion rate (LER) was extracted from time courses of leaf area in the platform and corresponded to the maximum first-

order derivative of P-spline fitted growth curves from 24 to 45 days at 20 °C after emergence[69]. Because this method provided somewhat unstable results, we also calculated maximum LER as the slope of the linear regression between leaf area and thermal time during the whole period from 24 to 45 days at 20 °C.

Genotypic values (BLUEs) for each trait were estimated by correcting raw traits values for spatial effects, by fitting a mixed model (R package SpATS[70],), with a fixed term for genotype and random effects for rows and columns as well as a smooth surface defined on row and column coordinates. Broad-sense heritabilities were calculated daily with the same R package, using the same model but with the genotype effect included as a random term. Regarding longitudinal traits, genotypic values at individual time points, t, were obtained from their smoothed time series using a generalized additive model fitted to the spatially adjusted daily measurements, $\tilde{y}_{i,k}(t)$, for each plant k of genotype i:

$$\tilde{y}_{i,k}(t) = \alpha_i + f_i(t) + \epsilon_{i,k}(t), \epsilon_{i,k}(t) \sim N(0, \sigma^2) \qquad (1)$$

where $\alpha_i$ is a genotype-specific intercept, $f_i(t)$ is a genotype-specific thin plate regression spline function on time, and $\epsilon_{i,k}(t)$ is a random error term (R package statgenHTP[69,71],).

Genomic heritability (narrow-sense, $h_g^2$) was estimated for each trait, panel and experiment with a model considering genomic-based additive and dominance relationship matrices[72], using the R package "BGLR"[73].

## Field experiments

The diversity panel was grown in 25 experiments (defined as combinations of site × year × water regime), either rainfed or irrigated, in ten sites in 2012 or 2013[31]. Sites were distributed on a west–east transect for temperature and evaporative demand, across Europe at latitudes from 44° to 49° N. The 'genetic progress' panel was grown in 26 field experiments either rainfed or irrigated, in 16 European sites from 2010 to 2017 spread along the same climatic transect as for experiments with the diversity panel[9]. The 'recent hybrids' panel was grown in four field experiments under irrigated conditions in the same range of latitudes, in 2021 or 2022 in France (Supplementary Table 2). Experiments followed an alpha-lattice design or randomized complete block design (RCBD) and were split by varieties maturity classes (Supplementary Table 2), each with three replicates of four-row plots, 6 m long. The targeted plant density was 9 plants m$^{-2}$. In all experiments, anthesis and silking dates were scored by visiting experiments every third day. The number of appeared leaves was scored every week on ten plants per hybrid during the vegetative phase, and leaf appearance rate was calculated as in indoor experiments (Supplementary Table 2).

The duration of the vegetative phase was defined as the period from plant emergence to anthesis, expressed in thermal time (equivalent days at 20 °C)[48]. Leaf appearance rate was estimated as in platform experiments.

UAV flights were performed three times in one experiment of the 'genetic progress' panel during the period from plant emergence to flowering, and seven times in two experiments for the 'recent hybrids' panel during the same period (Supplementary Table 2). Quadcopter drone (DJI Phantom 4) were equipped with a DJI multispectral camera with 5.7 mm focal length lens, acquiring 1600×1300 pixel images. They flew at a controlled altitude of 20 m and a constant speed of 2.2 m s$^{-1}$ for about 20 min per flight, with images captured at a one-second interval. Flights were performed during clear and cloudless days between 8:00 and 10:00 solar time. An automatic image-processing pipeline was applied by Hiphen, Avignon, France (http://www.hiphen-plant.com), following methods presented in Blancon et al.[20]. Environmental variables were recorded every hour in all experiments, including light, air temperature, relative humidity (RH), rainfall and wind speed. Soil water potential was measured every day with

tensiometers at 30 and 60 cm depths with three or two replicates, located in plots sown with a common reference hybrid.

The architectural trait ALA (Average Leaf inclination Angle to the soil level) and Leaf Area Index (LAI) were calculated by inversion of the PROSAIL model[41,74], based on multispectral images of field UAV flights. The PROSAIL model couples the PROSPECT leaf optical properties model with the SAIL canopy bidirectional reflectance model. It links the spectral variation of canopy reflectance, which is mainly related to leaf biochemical contents, with its directional variation, which is primarily related to canopy architecture and soil/vegetation contrast[75]. This link allows simultaneous estimation of canopy biophysical/structural variables from remote sensing, including ALA and LAI traits[42,76].

Leaf area index was also calculated by using a crop model (APSIM model as modified in Lacube et al.[11]) parameterized with the genotypic values (BLUEs) of four traits measured in PhenoArch platform (LAR, maximum leaf growth rate, response of leaf growth rate to VPD and final leaf number), plus the environmental and growing conditions recorded in the considered field.

Spatial corrections, calculations of genotypic values and heritabilities of traits were performed with the same methods as in indoor experiments.

## Correlation analysis between experiments

Pearson (r) and Spearman (rho) correlation coefficients were calculated to evaluate to which extent the genotypic values (BLUEs) of traits match between experiments, either in one field and another one, or in one field and the indoor platform. Both types of correlations was performed on the hybrids that were common to considered experiments (common hybrids number ranged from 18 to 56, Supplementary Data 1 and Supplementary Table 4). The significance of correlation coefficients was evaluated based on the null hypothesis that there is no correlation between the variables (r or rho = 0). Genetic correlations ($r_g$) between experiments were also assessed, using a multivariate Bayesian Gaussian mixed model, fitted for each couple of experiments (bivariate analysis, Table 2), with MTM R package[38,39]. Model fitting was based on 60,000 iterations, after discarding 10,000 cycles for burn-in period and using a thinning rate of 5. Each multivariate model implemented had the form:

$$\boldsymbol{y}_i = \mu_i \mathbf{1}_n + \boldsymbol{Z}_{ai} \boldsymbol{a}_i + \boldsymbol{Z}_{di} \boldsymbol{d}_i + \boldsymbol{\varepsilon}_i \qquad (2)$$

where the subscript $i$ refers to experiments (two experiments analyzed jointly, with trait values measured either indoor and in a field or in two different fields, Table 2), $y_i$ is the vector of trait values (BLUEs) of n hybrids in the considered couple of experiments, $\mu_i$ is the overall mean (intercept), $a_i$ is the vector of random additive genetic effects, $d_i$ is the vector of random dominance effects and $\varepsilon_i$ is the vector of random residual effects. $Z_{ai}$ and $Z_{di}$ are the incidence matrices for $a_i$ and $d_i$, respectively.

Variance components were calculated assuming: $a_i \sim$ MVN (0, $G_A \otimes V_a$) with $G_A$ as the genomic-based additive relationship matrix described below and $V_a$ as the additive effects variance–covariance matrix, $d_i \sim$ MVN (0, $G_D \otimes V_d$) with $G_D$ as the genomic-based dominance relationship matrix described below and $V_d$ as the dominance effects variance–covariance matrix, $\varepsilon_i \sim$ MVN (0, $I \otimes R$) where I denotes the identity matrix and R is the residual effects variance–covariance matrix.

Standard errors (SE) of all correlation coefficients were estimated using Bonett and Wright approximations[77]. Additionally, the theoretical accuracy of indirect selection (iAcc), i.e. in case of an indirect phenotypic selection based on observed values in a given experiment (indoor or in a field), was calculated as the genetic correlation between the considered couple of experiments, multiplied by the square root of trait genomic heritability in the reference experiment for selection[52] (Supplementary Table 4). Then, we quantified the efficiency of indirect selection relative to a direct phenotypic selection in the targeted

environment (Eff), by dividing the accuracy of indirect selection by the square root of trait genomic heritability in the target field experiment[52].

Root mean squared error of estimations (RMSE) and bias showing the discrepancy between experiments genotypic values (BLUEs) were calculated too. We present them as a coefficient of variation of the error ($CV_{RMSE}$) or bias ($CV_{Bias}$), which is the RMSE or bias expressed as a percentage of the mean value. Finally, to appreciate the consistency between experiments of the highest and lowest genotypic values, we evaluated the frequency of similar assignment to the highest or the lowest quartile between experiments for each trait. This consisted of estimating how many hybrids of the highest quartile of one experiment were also present in the highest quartile of the other experiment (Supplementary Table 4). The same was performed for the lowest quartile.

## Genotypic data and diversity analyses

All panels were genotyped using the 600 K Affymetrix® Axiom® array[78]. Genotypes of the hybrids were either inferred from genotypes of the parental lines (diversity panel) or resulted from the direct genotyping of the hybrids (genetic progress and recent hybrids panels). After quality control, 440,000 polymorphic SNPs were retained for diversity analyses and genomic prediction (excluding SNPs with minor allele frequency lower than 0.05 and/or missing values for more than 20% of hybrids). Missing values were otherwise imputed using BEAGLE v3[79].

Genotypic data generated were organized as M matrices with N rows and L columns, N and L being the panel size and number of markers, respectively. Genotype of hybrid n at locus (SNP marker) j was coded as 0 (the homozygote for B73 line allele), 1 (the heterozygote) or 2 (the other homozygote). "snpReady" R package[80] was used to estimate observed heterozygosity as

$$H_o = \frac{1}{L}\sum_{j=1}^{L}(nH_j/N) \tag{3}$$

and Nei's index of genetic diversity as

$$Nei\_GD = \frac{1}{L}/\sum_{j=1}^{L}(1-p_j^2-(1-p_j)^2) \tag{4}$$

with N the number of hybrids, $nH_j$ is the number of heterozygous hybrids at the jth biallelic locus, L is the total number of loci, and $p_j$ is the frequency of the reference (B73 line allele) at locus j (Supplementary Table 7). Principal Coordinate Analysis (PCoA) was also performed on SNP markers data (Supplementary Fig. 5).

## Genomic prediction model

Genomic predictions of each trait was performed with a genomic best linear unbiased prediction model (GBLUP-AD), including random additive and dominance effects:

$$y = \mu\mathbf{1}_n + Z_a a + Z_d d + \varepsilon \tag{5}$$

where y is the vector of trait genotypic means (BLUEs over experiments) of n hybrids, μ is the overall mean (intercept), $a$ is the vector of random additive genetic effects, and is assumed to follow a normal distribution $\sim N(0, G_A\sigma_a^2)$ with $G_A$ as the genomic-based additive relationship matrix described below and $\sigma_a^2$ as the additive genetic variance; d is the vector of random dominance effects which follows a normal distribution $\sim N(0, G_D\sigma_d^2)$ with $G_D$ as the genomic-based dominance relationship matrix described below and $\sigma_d^2$ as the dominance genetic variance; $\varepsilon \sim N(0, I\sigma_\varepsilon^2)$ is the vector of random residual effects, where I denotes the identity matrix and $\sigma_\varepsilon^2$ is the residual variance. $Z_a$ and $Z_d$ are the incidence matrices for $a$ and d, respectively.

The genomic-based relationship matrices were built as defined in Vitezica et al.[72] and González-Diéguez et al.[81]. The genomic-based

additive relationship matrix ($G_A$), called realized relationship matrix was estimated as

$$G_A = \frac{H_a H_a'}{tr(H_a H_a')/N} \tag{6}$$

where $H_a$ is a rescaled genotype matrix $H_a = M-P$, where M is the genotype matrix coded as 0, 1, and 2 for genotypes BB, Bb and bb respectively and with dimensions number of hybrids (N) by number of loci (L); P is the matrix of locus scores $2p_j$, with $p_j$ being the reference allele frequency of the $j^{th}$ SNP biallelic locus (having alleles B/b); tr is the trace. The genomic-based dominance relationship matrix ($G_D$) was estimated as

$$G_D = \frac{H_d H_d'}{tr(H_d H_d')/N} \tag{7}$$

where $H_d$ is the matrix containing elements $h_d$ for each individual and locus equal to:

$$h_d = \begin{cases} -2p_{Bb}p_{bb}\left[p_{BB}+p_{bb}-(p_{BB}-p_{bb})^2\right]^{-1} \\ 4p_{BB}p_{bb}\left[p_{BB}+p_{bb}-(p_{BB}-p_{bb})^2\right]^{-1} \\ -2p_{Bb}p_{BB}\left[p_{BB}+p_{bb}-(p_{BB}-p_{bb})^2\right]^{-1} \end{cases} \text{for genotypes} \begin{cases} BB \\ Bb \\ bb \end{cases} \tag{8}$$

where $p_{BB}$, $p_{Bb}$, and $p_{bb}$ are the genotypic frequencies for the genotypes BB, Bb, and bb respectively at the locus.

For moderate heritability physiological traits (LER and $gs_{max}$), in addition to random additive and dominance effects estimated from genomic-based relationship matrices, the genotypes of markers associated to quantitative trait loci (QTLs), previously identified in a genome-wide association study (GWAS) of the diversity panel[14], were added as fixed effects in prediction models:

$$y = \mu\mathbf{1}_n + X\beta + Z_a a + Z_d d + \varepsilon \tag{9}$$

where X is an n × l marker genotype matrix for n hybrids and l markers associated to trait QTLs and β is the markers fixed effects vector.

To test if G-BLUP model predictions are capturing genetic effects above that explained by population structure, we fitted a simple PC-BLUP model to the same data. In this model, the genotypes coordinates on the first five axes of SNP PCoA of the panels (representing of a cumulative percentage of variance of 35%), were used as predictors. Other genomic prediction models (RR-BLUP, BayesB, BayesC, and BayesR) were also tested but showed no significantly better results than those presented in this paper.

All prediction models were fitted using the Bayesian Generalized Linear Regression (BGLR) R package[73], based on 60,000 iterations, after discarding 10,000 cycles for burn-in period and using a thinning rate of 5.

## Training and validation schemes of genomic predictions

Genomic predictions were first evaluated by a 5-fold cross-validation scheme (CV1) repeated 10 times, applied to diversity and genetic progress panels datasets. In CV1, we aimed to measure the ability of the models to predict the performance of hybrids that would not have been evaluated in any of the observed environments[28]. For each iteration, the two panels genotypes were split into 5 subsets, then each subset (one fifth) was predicted using the remaining four fifths as training set. This generated a total of 5 × 10 testing sets. Each training set was sampled randomly but proportionally to 'diversity panel' genetic groups and across years of release of 'genetic progress' hybrids (Supplementary Fig. 4). This sampling method was chosen to maintain

a good coverage of the total genetic space covered by the training set[82]. The 'recent hybrids' panel dataset was then considered as an external validation of the prediction models. Here, the 'diversity' and 'genetic progress' panels were used together as training set, and predictions were made for the recent hybrids observed in independent experiments.

Five statistics were calculated to assess the performance of prediction models for each trait: the Pearson (r) and Spearman (rho) correlation coefficients between observed genotypic means (BLUEs over experiments) and predicted values (G-BLUPs), the prediction accuracy of genomic selection (Acc, estimated as the predictive ability (r) divided by the square root of trait genomic heritability[53]), the root mean squared error of predictions (RMSE) showing the discrepancy between predicted and observed values and the coefficient of variation of the error ($CV_{RMSE}$), which is the RMSE expressed as a percentage of mean observed value. For cross-validation scheme, the statistics estimations were performed within each fold and then averaged across folds and iterations. Standard errors (SE) of correlation coefficients were calculated using Bonett and Wright approximations[77].

### Reporting summary
Further information on research design is available in the Nature Portfolio Reporting Summary linked to this article.

## Data availability
The data generated in this study have been deposited in the Recherche Data Gouv database [https://recherche.data.gouv.fr/fr]. The datasets for phenotypic and genotypic values for the diversity panel are available at https://doi.org/10.15454/IASSTN. The datasets for phenotypic and genotypic values for the genetic progress panel are available at https://doi.org/10.15454/KLD0GH. The dataset for the 'recent hybrid' panel is available at https://doi.org/10.57745/NZY1KL. Source data are provided with this paper.

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

## Acknowledgements

This work was supported by the ANR projects ANR-10-BTBR-01 (Amaizing) and ANR-11-INBS-0012 (Phenome), by an ANRT CIFRE funding 2020/1238, and by INRAE and ARVALIS. Authors are grateful to M. Lis, R. Le Roy, B. Suard, and M. Combes for their technical assistance in the PhenoArch experiment, as well as to key persons from the 2021 and 2022 field experiments at ARVALIS, namely Y. Pousset, P. Raccurt, S. Bourrely, F. Binet, D. Lasserre, L. Pligot, C. Huet, M. Heredia, L. Diez, C. Drillaud, and B. Baron. We are also grateful to I. Granato, N. Abu-Samra Spencer, E. Millet, S. Prado, and C. Fournier for their help with data analysis or advice for R packages.

## Author contributions

J.B., L.M., C.W., M.B., F.T., and M.B. designed research. J.B. performed experiments, supervised by K.B., N.M., C.W., M.B., and F.T. Ll.C.B. and R.C. performed the experiments. J.B., L.M., C.W., and M.B. performed genomic analyses. J.B., C.W., and F.T. performed phenotypic and modeling analyses. F.T., J.B., C.W., L.M., and M.B. wrote the paper.

## Competing interests

The authors declare no competing interests.
