## [Peer Review File · Nature Communications]

Robotized indoor phenotyping allows genomic prediction of adaptive traits in the fieldReviewers' Comments:

Reviewer #1:

Remarks to the Author:

The manuscript by Boudghaghen et al. focuses on studying wheater physiological phenotypes that can be measured indoor can be used to breed for field traits, such as draught resistance.

The focus of the study is relevant: (i) there have been important advances in indoor high throughput phenotyping (I-HTP) that allows measuring traits that were difficult to measure in large scale in the past, (ii) in the past, changes in these traits, if it happened, was all based on indirect response to selection for agronomic traits. (ii) The combination of I-HTP with genomic selection, implemented in a rapid cycling scheme, has potential to change important aspects plant breeding.

The study then evaluates to what extent I-HTP phenotypes are correlated with similar measurements in the field, and evaluates genomic prediction accuracy for I-HTP.

Overall, the results seem promising and I believe that, if adequately revised, this manuscript will be valuable.

Major concerns

While I think that the questions are relevant and my feeling is that the results show promise, I was disappointed by several aspects of the methodology.

In the opening paragraph of the discussion the authors outline very well the framework one needs to follow to study this problem: (i) The indicator traits (I-HTP) should be heritable and genetically correlated with important field traits, and (ii) genomic prediction of I-HTP phenotypes should be accurate, making it feasible to implement rapid cycling based on indoor phenotyping.

The first condition (i) is based on standard selection theory, which goes back to Smith (1936) and Hazel (1943) and is covered in standard quantitative genetics books. However, this simple framework is not mentioned/reviewed and the key parameters, the accuracy of indirect selection (ACC) and the relative efficiency are not reported.

1) I think that the above framework should be outlined from the onset and should guide the methods and the presentation of the results. The current version, it somehow does, but in a rather ad-hoc manner. For instance: The estimates in Table 1 are correlations of blues which are not necessarily good estimates of genetic correlations (to do this, you should use a standard methods to estimate genetic correlations). Heritabilites, the other parameter affecting ACC, are mentioned by not fully reported in the main text. And, as noted, there are not reports of the accuracy of indirect selection and the relative efficiency (i.e., the ratio of the accuracy of indirect selection to the accuracy of direct selection). To me, the evidence will be stronger and the presentation clearer if you outline this framework and stick to report estimates of the parameters that are key to evaluate indirect selection.

2) I am not sure why are the correlation between phenotypes in different fields that relevant to your argument. I understand that the idea is to show that genetic correlations are below one even for the same trait measured in two different trials (e.g., different locations, different years) but I don't think this should occupy a central role relative to what I discussed in I.1

3) Genetic correlations, heritabilities, and even the prediction performance in CV are likely to be affected by the structure of the genotypes. The three data sets (particularly the diversity panel and the one based on hybrids that span over decades of release) are likely to have a level of genetic

diversity, and possibly structure, that is much higher than the one in current elite line pools. It would be important to: (i) assess population stratification, and (ii) possibly correct by it including in the models used to estimate heritabilities and genetic correlations a few PCs that can control for that.

4) Also, related to the above, are the correlations estimated within data set or using a pooled analysis where systematic differences between data sets contributes to variance and co-variance?

Not so central but important concerns

5) When you report estimates (e.g., correlations in Table 1, or in other places) you should report SE or CI. (Note that the correlations in Table 1, which are correlations between blues are not good estimates of genetic correlations).

6) For the CV, it is not clear whether you use a leave-genotype-out CV scheme (CV1 in Burgueño et al., Crop Sci, 2010) or one where individual data points are assigned to fold (CV2). These CV schemes represent very different prediction problems and would yield very different results. I suggest you do both.

7) I don't like the plot where you present the average prediction in CV, since the correlations are between BLUES and CV-predictions of them, you should present individual phenotype-predictions, not averages across CV in the plot. Also, it was not clear to me what do you mean by 'with 10 iterations' do you mean that you repeated the CV 10 times?

8) Many paragraphs begin with a dash, this is likely coming from early drafts, please be sure to review the manuscript carefully before resubmission.

9) In general, I think the writing can be improved significantly.

10) Try to produce subtitles and titles that are shorter and can be read and understood quickly.

11) Line 496, you write: "Genotypic values (BLUEs) for each trait were calculated by correcting raw traits values for spatial effects, by fitting.." Genetic values are unknown, the model estimate them. This, which may seem trivial, is highly relevant, we cannot calculate genetic values we can estimate them, and estimates are subject to estimation error. That is one of the reasons why we don't estimate genetic correlations by correlating predictions of genetic values. Instead, we used models where, in the marginal likelihood (or REML) genetic values are integrated out.

Reviewer #2:
Remarks to the Author:
Dear Authors,

I have finisher reading the manuscript. I think it is very interesting and useful for researchers working in this field.

Below you can find a few comments:

1) It will be interesting to include as an example the case of a trait that is measured indoors and in the field and whose correlation between genotypic values is close to zero (or even negative)

2) The notation in table 1 is confusing at least for me, can you replace "indoor vs field" in with the actual field number (e.g. 1, 2, 3, 4) given in the supplementary table 3?

3) Table 1, "correlation field vs field", can you use the actual field number?, e.g. 1,2,3,4.

4) Figure 1, "predicted values in a 5-fold cross validation scheme with 10 iterations", I think is better to write to say that you repeated the cross-validation experiment 10 times as you pointed out in line 598. Another alternative is to write that you performed a simulation experiment with 80% of data in training set and 20% in testing set and that you generated 50 of this partitions at random. This legend appears also in other figures.

5) You need to discuss about how the GxE interaction could affect your results.

6) For the genomic prediction model (2), why are you including the dominance effects? I assume that the "genotypic values" are derived using this model.

7) You need to use the conventions to write the linear models, the matrixes and vectors should be written in boldface fonts.

Regards.

Reviewer #3:

Remarks to the Author:

The manuscript by Boudighaghen et al. reports innovative research that explores new approaches to exploit key physiological traits measured by automatic phenotyping platforms under control conditions for the purpose of predicting adaptive traits in field environments. By modelling environmental effects, the team showed that the traits can be translated from indoors to fields and therefore this could be used in a range of ways to accelerate crop improvement for future environments. In particular, the genomic prediction application is very promising with high prediction accuracies reported for diverse maize hybrids and moderate prediction accuracies obtained for elite hybrid despite them showing a smaller phenotypic range of the traits of interest. By targeting key traits that determine the timing of water use, it could open the door for trait stacking or targeted trait selection for environments that require constrained water use such as drought environments or traits that enable the crop to maximise yield where water is not limiting. Over the years researchers have certainly doubted the ability to translate traits from controlled environment to field and this is understandable considering how different controlled environments are to field conditions. However, the authors in this study have targeted traits that can be used to predict similar traits in the field by accommodating the way in which the data is analysed and modelled. I think this is an important learning for researchers in the field and the plant breeding community who might be interested in utilising automated phenotyping platforms such as the one applied in this study. An extension of the research presented in this study would be exploring how the adaptive traits could add value to genomic predictions for yield. A multivariate or multi trait approach could be used to explore this if yield datasets are available, but not essential for this study.

Overall, the authors put forward extensive datasets demonstrating the utility of the physiological traits and how they correspond to related adaptive traits in the field and across environments. The article is clearly written and easy to follow. Below I have provided some minor comments and suggestions that the authors might like to consider in their revised version:

- I suggest removing the word 'genomic' from the title as the authors demonstrate an ability to predict adaptive traits from glasshouse to field could be based on both phenomic and genomic datasets.
- In the last sentence of the abstract, should be more specific and reference 'water-use' when referring to spender or conservative strategies.
- In Figure 4, predicted LAI values from indoors are higher than LAI predicted in the field using UAV. What might be driving this and what could be done to further improve the relationship? The authors

might like to comment in the discussion or results section.

- Lines 392-405: I agree testing genomic prediction on the recent hybrid varieties is a great thing to do and resembles what might be done in a breeding programme. It might be useful to calculate the genetic variance for the traits within the different groups of hybrids. This is different to the whole genome genetic diversity within the groups (shown in Supp Figure 3). This might reveal that genetic variance for the traits is generally lower in the group of recent hybrid varieties, not just the phenotypic range. Perhaps the authors could create a table summarising genetic variance for each of the traits, in each environment, for each of the groups.

Reviewer #4:

Remarks to the Author:

In the manuscript, authors attempted to use robotized indoor phenotyping techniques for some adaptive traits that will allow genomic prediction of those traits in the field. Certainly, the study has enough information and value for publishing in the journal. Though genomic prediction in unrelated lines were not that good but that is understandable as genomic prediction usually works better for validation panel which is related to training population. Using robotic technique for phenotyping of complex trait like stomatal conductance, would certainly be helpful for breeding program to develop stress tolerant corn genotype. One thing I was missing in the paper is the association between those adaptive traits with grain yield in corn. That information is important for the breeders. I suggest adding that in a table or in the result section.

REVIEWER COMMENTS

Reviewer #1 (Remarks to the Author):

The manuscript by Boudighaghen et al. focuses on studying whether physiological phenotypes that can be measured indoor can be used to breed for field traits, such as draught resistance.

The focus of the study is relevant: (i) there have been important advances in indoor high throughput phenotyping (I-HTP) that allows measuring traits that were difficult to measure in large scale in the past, (ii) in the past, changes in these traits, if it happened, was all based on indirect response to selection for agronomic traits. (ii) The combination of I-HTP with genomic selection, implemented in a rapid cycling scheme, has potential to change important aspects plant breeding.

The study then evaluates to what extent I-HTP phenotypes are correlated with similar measurements in the field, and evaluates genomic prediction accuracy for I-HTP.

Overall, the results seem promising and I believe that, if adequately revised, this manuscript will be valuable.

Major concerns

While I think that the questions are relevant and my feeling is that the results show promise, I was disappointed by several aspects of the methodology.

In the opening paragraph of the discussion, the authors outline very well the framework one needs to follow to study this problem: (i) The indicator traits (I-HTP) should be heritable and genetically correlated with important field traits, and (ii) genomic prediction of I-HTP phenotypes should be accurate, making it feasible to implement rapid cycling based on indoor phenotyping.

The first condition (i) is based on standard selection theory, which goes back to Smith (1936) and Hazel (1943) and is covered in standard quantitative genetics books. However, this simple framework is not mentioned/reviewed and the key parameters, the accuracy of indirect selection (ACC) and the relative efficiency are not reported.

1) I think that the above framework should be outlined from the onset and should guide the methods and the presentation of the results. The current version, it somehow does, but in a rather ad-hoc manner.

The corresponding paragraph of the introduction has been changed. Note that, as described in Methods section, the studied panels are not breeders' populations, so we did not carry out any selection scheme. What was done in this version is calculating the theoretical accuracy of indirect selection, in case of selection (i) based on observed values in different experiments (indoor and/or in different fields), or (ii) based on genomic prediction of trait values (prediction accuracy). The theoretical accuracy was estimated as the genetic correlation divided by the square root of trait heritability (Habier et al. 2010⁵², Dekkers et al. 2021⁵³, Table 2).

For instance: The estimates in Table 1 are correlations of BLUEs which are not necessarily good estimates of genetic correlations (to do this, you should use standard methods to estimate genetic correlations).

We have added in this version estimations of genetic correlations (Table 2), calculated using a multi-trait model (de los Campos et al. 2016³⁸, 2022³⁹), which takes into account genomic relationship matrices.

Heritabilities, the other parameter affecting ACC, are mentioned by not fully reported in the main text.

We have added a table summarising heritabilities and genetic variance components (Table 1, Supplementary Table 4).

And, as noted, there are not reports of the accuracy of indirect selection and the relative efficiency (i.e., the ratio of the accuracy of indirect selection to the accuracy of direct selection). To me, the evidence will be stronger and the presentation clearer if you outline this framework and stick to report estimates of the parameters that are key to evaluate indirect selection.

We have calculated in this version the theoretical accuracy of indirect selection (estimated as the genetic correlation divided by the square root of trait heritability, Habier et al. 2010⁵², Dekkers et al. 2021⁵³, Table 2).

2) I am not sure why are the correlation between phenotypes in different fields that relevant to your argument. I understand that the idea is to show that genetic correlations are below one even for the same trait measured in two different trials (e.g., different locations, different years) but I don't think this should occupy a central role relative to what I discussed in I.1.

Indoor measurements are often questioned for their relevance to field conditions, while the question of GxE in the field is considered as intrinsic to any field measurement. We feel essential to consider correlations between different fields as a benchmark for evaluating the translation of trait values from indoor to field. This is now explained in the text.

3) Genetic correlations, heritabilities, and even the prediction performance in CV are likely to be affected by the structure of the genotypes. The three data sets (particularly the diversity panel and the one based on hybrids that span over decades of release) are likely to have a level of genetic diversity, and possibly structure, that is much higher than the one in current elite line pools. It would be important to: (i) assess population stratification, and (ii) possibly correct by it including in the models used to estimate heritabilities and genetic correlations a few PCs that can control for that.

(i) The genetic structures of diversity panel and genetic progress panels were analyzed in previous papers (Negro et al. 2019⁶⁷ and Welcker et al. 2022⁹, respectively). We thought that presenting it again would distract attention. Note that in our study, all genetic analyses considered genomic-based additive and dominance relationship matrices (Vitezica et al. 2017⁷², González-Diéguez et al. 2021⁸⁰).

(ii) We now report heritabilities estimated for each panel (Table 1) and each experiment (Supp. Table 4). The correlations shown in Table 2 are also estimated within data of each couple of experiments (so not using a pooled analysis over panels).

We assumed that by using only genomic relationship matrices information, we were predicting both the structuration and relationship between genotypes:

- As described in Methods section, the experiments carried out for each panel were independent (no overlapping genotypes between them, so no practical possibility for correcting traits observed values for probable systematic differences between panels).*
- In the cross-validation scheme of genomic prediction, 246 out of 302 hybrids belonged to the same diversity panel and were phenotyped together. The remaining 56 hybrids belonged to the genetic progress panel and were observed in independent experiments.*
- In the external validation scheme of genomic prediction, the lower predictive ability may be due to systematic differences between the external panel (recent hybrids) and the two panels used as training population, in addition to other factors including panel size, genetic distance and less accurate estimation of genotypic values (less replicates or experiments carried out for the external panel).*

4) Also, related to the above, are the correlations estimated within data set or using a pooled analysis where systematic differences between data sets contributes to variance and co-variance? *As mentioned above, the correlations shown in Table 2 and Supp. Table 4 are estimated within dataset.*

Not so central but important concerns

5) When you report estimates (e.g., correlations in Table 1, or in other places) you should report SE or CI. (Note that the correlations in Table 1, which are correlations between BLUEs are not good estimates of genetic correlations).

We now report SE of correlations in Table 2 and Supplementary Tables 5-6, calculated as in Bonett et al. (2000)⁷⁶.

6) For the CV, it is not clear whether you use a leave-genotype-out CV scheme (CV1 in Burgueño et al., Crop Sci, 2010) or one where individual data points are assigned to fold (CV2). These CV schemes represent very different prediction problems and would yield very different results. I suggest you do both.

We have used a 5-fold cross-validation scheme (Hastie et al. 2009, Springer Series): every one fifth (60 hybrids in our case) is predicted with the remaining four fifths (242 hybrids). In presented figures, to avoid plotting 5 CV sub-plots for each trait, we plotted the whole 302 hybrids population dots together, but every 60 dots are predicted with the 242 other dots. The statistics reported in legends and Supp. Tables correspond to mean values over folds and iterations. We preferred using a k-fold CV scheme rather than a leave-one-out CV scheme, which does not mimic real cases that breeders face (Jarquín et al. 2017, The Plant Genome).

In our genomic prediction study, we predicted the mean trait genotypic values (over experiments).

7) I don't like the plot where you present the average prediction in CV, since the correlations are between BLUES and CV-predictions of them, you should present individual phenotype-predictions, not averages across CV in the plot. Also, it was not clear to me what do you mean by 'with 10 iterations' do you mean that you repeated the CV 10 times?

CV scheme was iterated 10 times, which means that random sampling was iterated in the same way. Hence, as we have 5 folds in each iteration, the reported statistics are mean values over 5x10 test sets. Many papers present CV results with boxplots, we preferred to show scatter plots that can be compared between all figures of a trait and to highlight that correlations are not linked to extreme dots or separated sub-groups.

We presented average predicted values because there was not a big decrease or increase in prediction quality from one fold to another or one iteration to another; For example, r ranged from 0.56 to 0.59 in the case of LAR. The CV statistics reported in legends and supplementary tables correspond to mean values over folds and iterations. Note that we have chosen a CV sampling method which maintains a good coverage of the total genetic space covered by the training set (Bustos-Korts et al. 2019)⁸². Actually, each training set was sampled randomly but proportionally to 'diversity panel' genetic groups and across years of release of 'genetic progress' hybrids (Supplementary Fig. 4).

8) Many paragraphs begin with a dash, this is likely coming from early drafts, please be sure to review the manuscript carefully before resubmission.

Corrected in the new version

9) In general, I think the writing can be improved significantly.

We hope that this version is better

10) Try to produce subtitles and titles that are shorter and can be read and understood quickly. *Corrected in the new version*

11) Line 496, you write: "Genotypic values (BLUEs) for each trait were calculated by correcting raw traits values for spatial effects, by fitting.." Genetic values are unknown, the model estimate them. This, which may seem trivial, is highly relevant, we cannot calculate genetic values we can estimate them, and estimates are subject to estimation error. That is one of the reasons why we don't estimate genetic correlations by correlating predictions of genetic values. Instead, we used models where, in the marginal likelihood (or REML) genetic values are integrated out.

Corrected in this version. As mentioned before, we have added estimations of genetic correlations (Table 2), calculated using a multi-trait model (de los Campos et al. 2016³⁸, 2022³⁹) which takes into account genomic relationship matrices.

Reviewer #2 (Remarks to the Author):

Dear Authors,

I have finished reading the manuscript. I think it is very interesting and useful for researchers working in this field.

Below you can find a few comments:

1) It will be interesting to include as an example the case of a trait that is measured indoors and in the field and whose correlation between genotypic values is close to zero (or even negative)

We have added a new supplementary figure 2, where we directly compared (i) LAI estimated from UAV imaging in a field with (ii) LAI calculated from values of leaf area measured indoor, multiplied by the plant density in the field. Both values were not correlated (slightly negative r , non-significant correlation). This was because environmental conditions and management practices were too different in the greenhouse and in the field. Conversely, the method proposed here considers traits upstream of LAI, measured indoor, and calculates LAI in the field based on the genotypic values of these upstream traits, via a crop model which takes into account management practices and environmental conditions in the considered field.

Note that Supplementary Figure 3 in this version, also presents a case with low correlation between indoor and field measurements for LAR trait, when time is not corrected for temperature.

Conversely, the match is now better in Supplementary Fig. 2 (now Supp. Fig. 1). The original figure compared results in two different panels because we did not have the different dates for ALA in Field 5 at the first submission. The new version of the figure now compares the same hybrids, which increases the resemblance of indoor and field data.

2) The notation in table 1 is confusing at least for me, can you replace "indoor vs field" in with the actual field number (e.g. 1, 2, 3, 4) given in the supplementary table 3?

The new version of Table 1 (now Table 2) brings this information

3) Table 1, "correlation field vs field", can you use the actual field number?, e.g. 1,2,3,4.

The new version of Table 1 (now Table 2) brings this information

4) Figure 1, "predicted values in a 5-fold cross validation scheme with 10 iterations", I think is better to write to say that you repeated the cross-validation experiment 10 times as you pointed out in line 598. Another alternative is to write that you performed a simulation experiment with 80% of data in training set and 20% in testing set and that you generated 50 of these partitions at random. This legend appears also in other figures.

The new version of Methods section is hopefully clearer.

5) You need to discuss about how the GxE interaction could affect your results.

The discussion section now explicitly mentions GxE interaction, with new sentences added to its second and third paragraphs.

6) For the genomic prediction model (2), why are you including the dominance effects?

I assume that the "genotypic values" are derived using this model.

We included them because the variance component linked to dominance relationship matrix was appreciable for most studied traits (ranging from 14% to 50% of total genetic variance, Table 1). It is now stated in Methods section, that the genetic correlations reported in Table 2 are derived using a model considering both additive and dominance relationship matrices.

7) You need to use the conventions to write the linear models, the matrixes and vectors should be written in boldface fonts. *The new version takes these conventions into account.*

Reviewer #3 (Remarks to the Author):

The manuscript by Boudghaghen et al. reports innovative research that explores new approaches to exploit key physiological traits measured by automatic phenotyping platforms under control conditions for predicting adaptive traits in field environments. By modelling environmental effects, the team showed that the traits can be translated from indoors to fields and therefore this could be used in a range of ways to accelerate crop improvement for future environments. In particular, the genomic prediction application is very promising with high prediction accuracies reported for diverse maize hybrids and moderate prediction accuracies obtained for elite hybrid despite them showing a smaller phenotypic range of the traits of interest. By targeting key traits that determine the timing of water use, it could open the door for trait stacking or targeted trait selection for environments that require constrained water use such as drought environments or traits that enable the crop to maximise yield where water is not limiting. Over the years, researchers have certainly doubted the ability to translate traits from controlled environment to field and this is understandable considering how different controlled environments are to field conditions. However, the authors in this study have targeted traits that can be used to predict similar traits in the field by accommodating the way in which the data is analysed and modelled. I think this is an important learning for researchers in the field and the plant breeding community who might be interested in utilising automated phenotyping platforms such as the one applied in this study. An extension of the research presented in this study would be exploring how the adaptive traits could add value to genomic predictions for yield. A multivariate or multi trait approach could be used to explore this if yield datasets are available, but not essential for this study.

Overall, the authors put forward extensive datasets demonstrating the utility of the physiological traits and how they correspond to related adaptive traits in the field and across

environments. The article is clearly written and easy to follow. Below I have provided some minor comments and suggestions that the authors might like to consider in their revised version:

- I suggest removing the word 'genomic' from the title as the authors demonstrate an ability to predict adaptive traits from glasshouse to field could be based on both phenomic and genomic datasets. *If agreed by the reviewer and editor, we would prefer to keep the term 'genomic prediction', because this is an essential information that traits can be reasonably predicted based on genomic information, and not only correlated between indoor and field.*

- In the last sentence of the abstract, should be more specific and reference 'water-use' when referring to spender or conservative strategies. *This sentence was changed.*

- In Figure 4, predicted LAI values from indoors are higher than LAI predicted in the field using UAV. What might be driving this and what could be done to further improve the relationship? The authors might like to comment in the discussion or results section.

The former Fig. 4 considered the maximum LAI, simulated with an old version of the crop model (Lacube et al. 2020)¹¹ that did not sufficiently take senescence into account. The new version of the crop model results in lower LAI at flowering time via the senescence of first leaves.

Even so, there is still a saturation of PROSAIL at LAI of 4.5 in the new Fig. 4. A sentence was added to explain this. The PROSAIL inversion provides LAI that saturates when light interceptions becomes close to 100%.

- Lines 392-405: I agree testing genomic prediction on the recent hybrid varieties is a great thing to do and resembles what might be done in a breeding programme. It might be useful to calculate the genetic variance for the traits within the different groups of hybrids. This is different to the whole genome genetic diversity within the groups (shown in Supp. Figure 3). This might reveal that genetic variance for the traits is generally lower in the group of recent hybrid varieties, not just the phenotypic range. Perhaps the authors could create a table summarising genetic variance for each of the traits, in each environment, for each of the groups.

We have added a new Table 1, which summarises heritabilities and genetic variance components for each trait in the three studied panels. We also have added a new Supplementary Table 4 summarising them in all reported experiments.

Reviewer #4 (Remarks to the Author):

In the manuscript, authors attempted to use robotized indoor phenotyping techniques for some adaptive traits that will allow genomic prediction of those traits in the field. Certainly, the study has enough information and value for publishing in the journal. Though genomic prediction in unrelated lines were not that good but that is understandable as genomic prediction usually works better for validation panel which is related to training population. Using robotic technique for phenotyping of complex trait like stomatal conductance would certainly be helpful for breeding program to develop stress tolerant corn genotype. One thing

I was missing in the paper is the association between those adaptive traits with grain yield in corn. That information is important for the breeders. I suggest adding that in a table or in the result section.

We agree that association between adaptive traits with grain yield is of major importance. We are currently carrying out this study, with the following results

- Correlations between yield and traits are obscured by confusion of effects with flowering time: we first needed to take out the effect of flowering time before considering correlations.
- As expected for adaptive traits, their association with grain yield depended on environmental conditions in the considered field. For instance, stomatal conductance was positively associated with grain yield in well-watered fields with a cool temperature, and negatively correlated with it in dry fields (see below table). The same applied for nearly all studied traits.
- For demonstrating this, we first needed to cluster field into environmental scenarios, and then to consider the association of each trait to grain yield in each scenario.
- For being convincing, we considered genetic correlations, by taking SNP markers effects into account, and not only phenotypic correlations.

Overall, results showed that adaptive traits were genetically correlated with grain yield, but after a non-trivial data analysis that merits to be evaluated per se by reviewers. The current MS would largely exceed the word limit and number of figures if we tried to place this information in it. Hence, we would prefer to submit this paper per se (some work remains to be done), and avoid placing the yield analysis in the current MS. We still added a paragraph on this topic in the discussion section of the current MS.

Yield per environmental scenario		'Constitutive' Traits in platform		'Adaptive'				
		Anthesis	Architecture RH _{PAD}	Stomatal conductance	Light interception	Leaf area	WUE	Sensitivity growth
	WW Cool	Green	Light Green	Green	Light Green	Light Green	Light Pink	Orange
	WW high VPD	Green	Light Green	Green	Light Green	Light Green	Light Pink	Orange
	WW Hot night	Green	Light Green	White	Light Pink	White	Light Green	Light Green
	WD Cool	Green	Light Green	Light Green	Light Pink	White	Green	White
	WD Hot	Green	Light Green	Orange	Light Pink	Light Pink	Green	White

Diversity panel (DROPS)

Reviewers' Comments:

Reviewer #1:

Remarks to the Author:

I will provide my comments in a pdf, because my review includes equations that I typed in word.

The authors have addressed most of my comments.

I am satisfied with the answers they provide to most of my comments.

I have only one major concern I still have are the following (numbering follows the numbers in my original review).

1) In their response, they state:

“The theoretical accuracy was estimated as the genetic correlation divided by the square root of trait heritability”

I think you are miss-interpreting the reference (Habier and Dekkers)

The selection accuracy (direct or indirect) is the correlation between the index used to rank individuals (I) and the true genetic merit for the selection objective (g), that is

$$Acc = Cor(I, g)$$

This can be shown to be equal to the product of the genetic correlation between I and g , times the square-root of the heritability of I , thus

$$Acc = \sigma_{G(I,g)} \sqrt{h_I^2}$$

Above, $\sigma_{G(I,g)}$ is the genetic correlation between the measured trait (I) and the selection objective.

When individuals are ranked based on a single phenotypic measurement of the trait of interest (direct, mass selection), the above formula becomes

$$Acc - Direct = \sqrt{h_y^2}$$

Above, h_y^2 is the heritability of the selection objective

Some authors (including the references you provide) express the prediction correlations relative to the accuracy of direct selection (correlation/sqrt(trait-heritability)), but this is the relative accuracy, not the accuracy of indirect selection.

So, please revise the formulas you have used for the accuracy of indirect selection. For references to this I suggest Falconer & Mackay or, a more recent paper by Lopez-Cruz et al., Scientific Reports 2020.

3) Response to my point # 3 of the previous review.

I was interested on them providing evidence that the prediction correlations they obtain are significantly higher than what can be explained by population structure. I am not satisfied with their response. Yes, genomic models account for population structure, however, my point is, if instead of using the models you use, you fit a simple model with the top 5 SNP-derived PCs, what level of correlation do you obtain? If those correlations are much lower than the ones you are reporting, that would provide evidence that the predictions you are deriving are capturing patterns above and beyond of what can be explained by population structure.

6) For the CV, it is not clear whether you use a leave-genotype-out CV scheme (CV1 in Burgueño et al., Crop Sci, 2010) or one where individual data points are assigned to fold (CV2). These CV schemes represent very different prediction problems and would yield very different results. I suggest you do both.

I am also not satisfied with this response. Even if you do a 5-fold CV, one can assign either individual records or genotype IDs to folds. This generates very different prediction problems. If you assign individual records to folds, then, if your data set includes multiple records per genotype, you will have records from the same genotype in training and testing (this is similar to what Burgueño et al 2010 refers as to CV2). On the other hand, if you assign genotype IDs to folds, that generates a scheme where no data from the genotypes in the testing fold are present in the training folds (this is similar to what Burgueño et al 2010 refers as to CV1).

I insist with this because it is impossible to evaluate prediction correlations from CV in absence of clarity how data (individual records, all the records from each genotype, families, etc) was assigned to folds.

Reviewer #2:
Remarks to the Author:
Dear authors,

I have read the new version of the manuscript and the responses to my comments and I think that you have addressed all of them successfully.

Regards.

Reviewer #3:
Remarks to the Author:
In this revised version, the authors have fully incorporated or addressed all of my suggestions or issues raised.

Response to reviewers

We only respond to reviewer 1, because reviewers 2 and 3 did not raise new points.

In addition, because the data analysis is progressing, we could add a second plot to Figure 4 corresponding to a water deficit treatment. This does not change the general message, nor the methods, but extend our conclusion to a larger range of environmental conditions.

Finally, the “data availability” statement was corrected, with the link to the phenotypic data (one dataset has been created for the ‘recent genotypes’ panel, <https://doi.org/10.57745/NZY1KL>

Reviewer 1

The authors have addressed most of my comments.

I am satisfied with the answers they provide to most of my comments.

I have only one major concern I still have are the following (numbering follows the numbers in my original review).

1) In their response, they state:

“The theoretical accuracy was estimated as the genetic correlation divided by the square root of trait heritability”

I think you are miss-interpreting the reference (Habier and Dekkers).

The selection accuracy (direct or indirect) is the correlation between the index used to rank individuals (I) and the true genetic merit for the selection objective (g), that is :

$$Acc = Cor(I, g)$$

This can be shown to be equal to the product of the genetic correlation between I and g , times the square-root of the heritability of I , thus :

$$Acc = \sigma_{G(I,g)} \sqrt{h_I^2}$$

Above, (I,g) is the genetic correlation between the measured trait (I) and the selection objective.

When individuals are ranked based on a single phenotypic measurement of the trait of interest (direct, mass selection), the above formula becomes :

$$Acc - Direct = \sqrt{h_y^2}$$

Above, h_y^2 is the heritability of the selection objective.

Some authors (including the references you provide) express the prediction correlations relative to the accuracy of direct selection (correlation/sqrt(trait-heritability)), but this is the relative accuracy, not the accuracy of indirect selection.

So, please revise the formulas you have used for the accuracy of indirect selection. For

references to this I suggest Falconer & Mackay or, a more recent paper by Lopez-Cruz et al., Scientific Reports 2020.

Thank you for the clarifications, reviewer 1 was right. In addition to performing the suggested calculation, we proposed that two different cases can be considered:

- The first case corresponds to the reviewer's suggestion. It considers a selection that would be purely phenotypic (first set of columns in Table 2). Here, we computed the accuracy of indirect selection (iAcc), as the genetic correlation between traits (either platform vs field or field vs field), multiplied by the square root of heritability of the trait in the reference experiment used to select. Then, as in Lopez-Cruz et al. 2020, we quantified the efficiency of indirect selection relative to direct phenotypic selection (Eff), by dividing the accuracy of indirect selection by the square root of trait heritability in the target field experiment. In Table 2 of the revised version, we reported these efficiencies of indirect selection values (Eff). In Supplementary Table 5, we reported the corresponding accuracies of indirect selection values (iAcc).
- The second case corresponds to our earlier versions, and was maintained in this revised version for genomic selection (GS). Here, as recommended for GS studies by several authors such as Lorenzana & Bernardo 2009 (DOI 10.1007/s00122-009-1166-3), the accuracy of genomic selection (Acc), was calculated as the correlation coeff (r) between observed and predicted values (also called predictive ability), divided by the square root of trait heritability. Indeed, in this case, one does not have access to the correlation between predicted genetic values and the true genetic values of the trait (true breeding values are unknown). GS accuracy (Acc) can be further compared to the accuracy of a direct phenotypic selection (by dividing Acc by the sqrt of trait heritability in the target environment). In Table 2, we preferred to report prediction accuracies (Acc) for genomic selection results part, as commonly published by other authors for GS studies.

3) Response to my point # 3 of the previous review.

I was interested on them providing evidence that the prediction correlations they obtain are significantly higher than what can be explained by population structure. I am not satisfied with their response. Yes, genomic models account for population structure, however, my point is, if instead of using the models you use, you fit a simple model with the top 5 SNP-derived PCs, what level of correlation do you obtain? If those correlations are much lower than the ones you are reporting, that would provide evidence that the predictions you are deriving are capturing patterns above and beyond of what can be explained by population structure.

We performed the calculations recommended by Rev 1, with a model taking into account the first five axes of the SNP PCoA, for all traits and validation schemes (Supp. Fig. 6, Supp. Table 7). The quality of prediction dramatically decreased for the external validation set. For the cross-validation scheme, the quality of prediction was also lower than with the G-BLUP model, because the overall correlation mainly resulted from separate clouds of points corresponding to each panel. This suggests that our G-BLUP model predictions captured genetic effects beyond that explained by

population structure.

6) For the CV, it is not clear whether you use a leave-genotype-out CV scheme (CV1 in Burgueño et al., Crop Sci, 2010) or one where individual data points are assigned to fold (CV2). These CV schemes represent very different prediction problems and would yield very different results. I suggest you do both.

I am also not satisfied with this response. Even if you do a 5-fold CV, one can assign either individual records or genotype IDs to folds. This generates very different prediction problems. If you assign individual records to folds, then, if your data set includes multiple records per genotype, you will have records from the same genotype in training and testing (this is similar to what Burgueño et al 2010 refers as to CV2). On the other hand, if you assign genotype IDs to folds, that generates a scheme where no data from the genotypes in the testing fold are present in the training folds (this is similar to what Burgueño et al 2010 refers as to CV1).

I insist with this because it is impossible to evaluate prediction correlations from CV in absence of clarity how data (individual records, all the records from each genotype, families, etc) was assigned to folds.

Actually, we performed the CV1 scheme, assigning genotype IDs to folds (we predicted mean genotypic values over experiments, for hybrids that would not have been evaluated in any of the observed environments). This is now clarified in the Methods section. Indeed, we considered that CV1 corresponds to our objective (predicting the performance of a new genotype) and is more challenging than CV2; so if the prediction accuracies are good with CV1, they would also be good for CV2 (cf. Jarquín et al. 2017 The Plant Genome, doi: 10.3835/plantgenome2016.12.0130). Additionally, our external validation (with recent hybrids panel) was a case of CV00 scheme, the most challenging situation, where we tried to predict the performance of new genotypes evaluated in independent new experiments.

Reviewers' Comments:

Reviewer #1:

Remarks to the Author:

The revision has addressed all the comments that I had.